# Field-based molecular detection of *Batrachochytrium dendrobatidis* in critically endangered *Atelopus* toads and aquatic habitats in Ecuador

Lenin R. Riascos-Flores[1,2], Julio Bonilla[2], Leopoldo Naranjo-Briceño[3,4], Katherine Apunte-Ramos[5], Grace C. Reyes-Ortega[6], Marcela Cabrera[7,8], José F. Cáceres-Andrade[9], Andrea Carrera-Gonzalez[5], Jomira K. Yánez-Galarza[3,6], Fausto Siavichay Pesántez[10], Luis A. Oyagata-Cachimuel[11], Peter Goethals[1], Jorge Celi[12], Christine Van der Heyden[13], H. Mauricio Ortega-Andrade[6,11]*

1 Department of Animal Sciences and Aquatic Ecology, Faculty of Bioscience Engineering, Gent University, Gent, Belgium, 2 Escuela Superior Politécnica del Litoral, ESPOL/Facultad de Ciencias de la Vida, Centro de Investigaciones Biotecnológicas del Ecuador, Campus Gustavo Galindo Guayaquil, Guayaquil, Ecuador, 3 Applied Microbiology Research Group, Life Sciences Faculty, Universidad Regional Amazónica IKIAM, Tena, Ecuador, 4 Fungal Biotech Lab, Spora Biotech, Huechuraba, Santiago, Región Metropolitana, Chile, 5 Molecular Biology and Biochemistry Lab, Universidad Regional Amazónica IKIAM, Tena, Ecuador, 6 Biogeography and Spatial Ecology Research Group, Life Sciences Faculty, Universidad Regional Amazónica IKIAM, Tena, Ecuador, 7 Laboratorio Nacional de Referencia del Agua, Universidad Regional Amazónica Ikiam, Tena, Ecuador, 8 University of Valencia, Paterna, Spain, 9 Parque Nacional Cajas ETAPA EP, Cuenca, Ecuador, 10 Centro de Conservación de Anfibios AMARU, Cuenca, Ecuador, 11 Herpetology Division, Instituto Nacional de Biodiversidad, Quito, Pichincha, Ecuador, 12 Water and Aquatic Resources Research Group, School of Earth and Water Sciences, Universidad Regional Amazónica Ikiam, Tena, Ecuador, 13 Department of Biosciences and Industrial Technology, Health and Water Technology Research Centre, HOGENT—University of Applied Sciences and Arts, Gent, Belgium

* mauricio.ortega@ikiam.edu.ec

**Data Availability Statement:** All relevant data are within the manuscript and its Supporting Information files.

## Abstract

*Batrachochytrium dendrobatidis* (Bd) is a lethal fungal species that parasitizes vertebrates and is associated with the worldwide decline of amphibian populations. The development of sensitive, rapid detection methods, particularly DNA-based techniques, is critical for effective management strategies. This study evaluates the efficacy of DNA extraction and a portable PCR device in a mountable field laboratory setup for detecting Bd near the habitats of three critically endangered *Atelopus* toad species in Ecuador. We collected skin swabs from *Atelopus balios*, *A. nanay*, *and A. bomolochos*, and environmental DNA (eDNA) samples from streams in Andean and coastal regions of Ecuador. For eDNA, a comparison was made with duplicates of the samples that were processed in the field and in a standard university laboratory. Our findings revealed Bd detection in eDNA and swabs from 6 of 12 water samples and 10 of 12 amphibian swab samples. The eDNA results obtained in the field laboratory were concordant with those obtained under campus laboratory conditions. These findings highlight the potential of field DNA-based monitoring techniques for detecting Bd in amphibian populations and their aquatic habitats, particularly in remote areas. Furthermore, this research aligns with the National Action Plan for the Conservation of Ecuadorian Amphibians and contributes to the global effort to control this invasive and deadly fungus.

**Funding:** CVdH: "Networks 2019 phase 2 Ecuador biodiversity project", the VLIR-UOS South Initiative ZEIN 2014Z15. CVdH, JC, JB, HMOA: "DNA-based monitoring for assessing the effect of invasive species on aquatic communities in the Amazon basin of Ecuador", the VLIR-UOS South Initiative EC 2020 SIN 277B125 "Biomonitoring of aquatic environments in the Amazon using environmental DNA" HMOA: "Conservation of Ecuadorian Amphibians and access to genetic resources-PARG" managed by the Ministry of Environment and Water of Ecuador; Project "On the quest of the golden fleece in Amazonia: The first herpetological DNA - barcoding expedition to unexplored areas on the Napo watershed, Ecuador", funded by the Secretaría Nacional de Ciencia y Tecnología del Ecuador (Senescyt-ENSAMBLE Grant #PIC-17-BENS-001), The World Academy of Sciences (TWAS Grant #16-095)", Erasmus+ CBHE consortium "Nature-based living-lab for interdisciplinary practical and research semester on sustainable development and environmental protection in the Amazona Rainforest [NB-LAB]" (Grant number: 619346-EPP-12020-1-DE-EPPKA2-CBHE-JP). The funders had no role in study design, data collection and analysis, decision to publish, or preparation of the manuscript.

**Competing interests:** The authors have declared that no competing interests exist.

## Introduction

The emerging disease chytridiomycosis, caused by the pathogen *Batrachochytrium dendrobatidis* (Bd), is a disease that has affected the three orders of amphibians (anura, urodela, and gymnophiona). This diseases continues to threaten these population, mainly due to expansion of human activity [1]. After a massive mortality event of frogs in Australia and Central America [2] and the reports of extensive propagation worldwide, Bd has been reported in over 700 species of amphibians [3, 4]. This pathogen is implicated in the decline of about 501 amphibian species and is linked to 90 possible extinctions worldwide [5, 6]. The highest rates of amphibian mortality and extinction attributed to Bd have predominantly occurred in the Americas and Oceania [7]. In South America, a significant mass die-off event occurred in the mid-2000s [5, 8]. This event profoundly impacted Ecuador, leading to the recognition of Bd as a major threat to local amphibian populations with *Atelopus* toads among the species most severely affected [9].

Bd transmission occurs through animal contact or via waterborne motile zoospores [10]. Detection of Bd in amphibian hosts, including larvae, metamorphs, and adults, is achieved through various methods. These include visual inspection of tadpoles, fungal culture, light microscopy, immune histochemistry, electron microscopy, immuno-electron microscopy, and polymerase chain reaction (PCR) [11, 12]. However, with the advances in genetics and genomic technologies, molecular techniques (PCR and qPCR) have become the most common and reliable methods for Bd identification [13–15].

*Atelopus* toads (Anura: Bufonidae) represent one of the most threatened amphibian groups worldwide, with at least 80% of their species facing the risk of extinction [16]. In Ecuador, this risk escalates to 100% for the 25 species identified [9]. In response, the Environmental Ministry of Ecuador initiated a national program in 2015, named project PARG, dedicated to the conservation of Ecuadorian amphibian biodiversity and the sustainable use of genetic resources. This program includes biomonitoring of key frog species in the country [17]. Among these are three Atelopus species, classified as critically endangered: *A. nanay* and *A. bomolochos*, endemic to the Central Andes highlands, and *A. balios*, endemic to the coastal region of the country [18]. The habitats of these species have been severely degraded by the cumulative impacts of human activities, climate change, and invasive species [9, 19, 20].

Effective control and management of Bd infections worldwide require comprehensive monitoring, including abiotic factors, such as physicochemical and environmental conditions, and biotic elements, including ecological interactions and species niches [21, 22]. Nonetheless, in developing countries, monitoring efforts often encounter significant challenges despite the regions' rich biodiversity and numerous species susceptible to Bd [16, 23]. These challenges include the high cost of equipment, limited access to well-equipped laboratory facilities, a scarcity of reagents, and complex bureaucratic procedures for importing materials [24, 25]. In this context, utilizing portable and low-cost sequencing devices for pathogen surveillance has emerged as a practical approach to assess biological diversity in global biodiversity hotspots [26–29]. Consequently, the primary aim of this study is to assess the reliability and accuracy of a field laboratory in detecting Bd within native populations of three endangered Atelopus toad species in the Central Andean highlands and coastal regions of Ecuador.

## Materials and methods

### Ethical statement

The Ministry of Environment of Ecuador provided the research permits "MAE-DNB-CM-2017-0062" and "MAATE-DBI-CM-2021-0177". No specimens were collected in this study.

### Study area

For this study, several factors were considered in selecting locations and species: Firstly, in the last five years, reports have indicated the persistence of Bd in isolated populations in the southern Andes of Ecuador [30, 31]. This influenced the choice of species, which was further limited by the small remaining populations. For instance, *A. bomolochos*, once thought to be extinct, was rediscovered in 2015 when a few individuals were sighted [32]. Another consideration was the sampling area limitations; for example, *A. bomolochos* and *A. nanay* are confined to the highlands of Cajas National Park in Azuay, Ecuador. These species face imminent extinction risks, with their habitats restricted to approximately 60 km$^2$ [9]. Finally, human interactions were taken into account; since 2015, the remaining individuals of the *A. Balios* specie in the "El Naranjal" area have been living in close proximity to agricultural activities [17].

During the monitoring phase, field laboratories were established near the monitoring sites. These labs, were designed to replicate the capabilities of a standard laboratory for sample processing and analysis. They were equipped with portable instruments such as PCR machines, centrifuges, vortex mixers, and electrophoresis chambers. The establishment of these field laboratories was driven by a dual purpose: firstly, to minimize the distances required for sample transportation, thereby reducing the risk of DNA degradation; and secondly, to enable the replication of laboratory conditions in remote and isolated locations, far from conventional research centers or universities.

Specific sampling sites were selected based on prior reports on the distributional range of three *Atelopus* species [9, 18] and the biomonitoring sites established by project PARG [17]. The travel time taken between the sampling sites and the field laboratory ranged from 2 to 4 hours. In contrast, transporting these samples to the university campus for processing took an additional two to three days by road.

Skin swabs and water samples were collected between 4-8th October 2020 in two localities from Azuay province: Cerro Negro (CN), 3.15675˚ S, 78.84538˚ W, 3064 m.a.s.l. (4 sites, *Atelopus bomolochos*), Sig Sig–Angas (A), Parque Nacional Cajas, 2.88337˚ S, 79.30685˚ W, 3800 m.a.s.l. (4 sites; *Atelopus nanay*); and two localities from Guayas province for *Atelopus balios*: Cerro de Hayas (CH), 2.72452˚ S, 79.61892˚ W, 107 m a.s.l. (2 sites), Comunidad San Miguel, Estero Arenas (EA), 2.75077˚ S, 79.01269˚ W, 203 m.a.s.l. (4 sites; See S1 Table for specific site description). Each site was located, at least, 100 m apart of distance.

### Amphibian sampling

Due to the limited number of adult specimens in the sampled areas [30, 31], amphibian sampling required a collaborative effort by a team of four to seven people. We followed biosafety protocols for handling specimens [33]. Specimens were captured by hand in the field using new nitrile gloves for each specimen captured. Skin sampling was performed using a sterile citoswab series® microbiological collection and transport system (Amies Charcoal Gel 2120–0025) in accordance with the protocols established by Angulo et al. [33].

During sampling specimens were sprayed with sterile MilliQ water. Swabbing was performed in skin ares–belly, groin, legs, feet, and hands–using a single swab for each specimen [33]. The swab was then stored in a 1.5mL microtube containing 400μL of lysis buffer (Tris-

HCl 0.18 M; EDTA 10 mM, SDS 1%, pH 8.2) and maintained in a cooler box. After sampling, specimens were released at the same place of capture. No specimens were euthanized.

## Water collection

Stream sampling methods in this study were based on those described by Quilumbaquin et al. [28]. Before arriving at the sampling sites and starting eDNA sampling, all materials were disinfected in the laboratory using a sodium hypochlorite solution (1/5 of the commercial solution), following the protocol from Geerts *et al.* [34]. The disinfection process included filtration systems and sampling bottles used at each site. To assess potential contamination, a blank sample (sterilized distilled water) was processed for each location. Field materials such as scissors and forceps were sterilized by rinsing with 96% ethanol followed by flame sterilization.

At each site, one liter of water was collected, dividing it into three approximately 350 ml samples taken from three points 10 meters appart. The water sample were then filtered over a 0.45 μm pore-sized nitrocellulose filter of 47 mm diameter (Porafil®). The filtering process utilized a Gast DOA-P704-AA High-Capacity Vacuum Pump and sterilized Nalgene filtration devices (ThermoFisher Scientific). To avoid cross contamination, 1 L of sterilized distilled water was filtered as a negative control before processing the samples. This process was repeated for each location.

Before extracting eDNA in the field-lab, filters were cut in half. One half was used to extract DNA in the field using portable devices, while the other half was stored at -20°C for later extraction in the university laboratory. Each half of the filter was cut into small 1 to 2 mm strips. These strips were handled with sterilized tools and were stored in sterile 1.5 ml Eppendorf tubes. Eppendorf tubes were cooled or frozen for field and university laboratories, respectively. Tools were sterilized between filtering steps by rinsing with 96% ethanol, and flame sterilization.

## DNA extraction from skin swabs and water samples

The detailed protocols for collecting, extracting and amplifying DNA from water samples and skin swabs for the detection of Bd using field devices are described in S1 File and in previous studies [34, 35]. During laboratory work and sample preparation, specifically for skin swabs, disposable gloves were worn and replaced between each sample to prevent cross-contamination. To prevent contamination in the field laboratory, a dedicated table was allocated for the preparation of the PCR master mix. Additionally, gel electrophoresis and PCR thermal cycling were performed in separate areas (Fig 1).

Extraction of fungal DNA from skin swabs was performed using and adapted version from the Wizard® Genomic DNA Purification Kit (Promega Cat. # A1120). The DNA extraction protocol was performed as follows: Swabs were mixed in 300 μL of Lysis buffer (Tris-HCl 0.18 M; EDTA 10 mM; SDS 1%, pH 8.2). Then, the swabs were removed using sterile tweezers. 10μL of Proteinase K (50μg/μL) (Invitrogen Cat. # QS0511) was added. After mixing by vortexing for 20 s at maximum speed (Vortex V-1 plus), the tubes were incubated on a thermoblock at 65°C for 30 min. Then, 2.5 μL RNase (10 mg/mL) (Thermo Scientific, Cat. # EN0531) was added and mixed by pipetting before incubation at 37°C for 10 min.

For cells and protein precipitation, 100μL of protein precipitation solution (Promega) was added and vortexed for 20s, followed by incubation for 10 min on ice. After centrifugation at 12000 rpm for 5 min, the supernatant was recovered and transferred to a clean sterilized 1.5 mL microtube. To precipitate the DNA, 40 μL of Sodium acetate (3M, pH 5.2) and 1000μL of absolute ethanol were added and centrifuged for 5 min at 12000rpm. The supernatant was

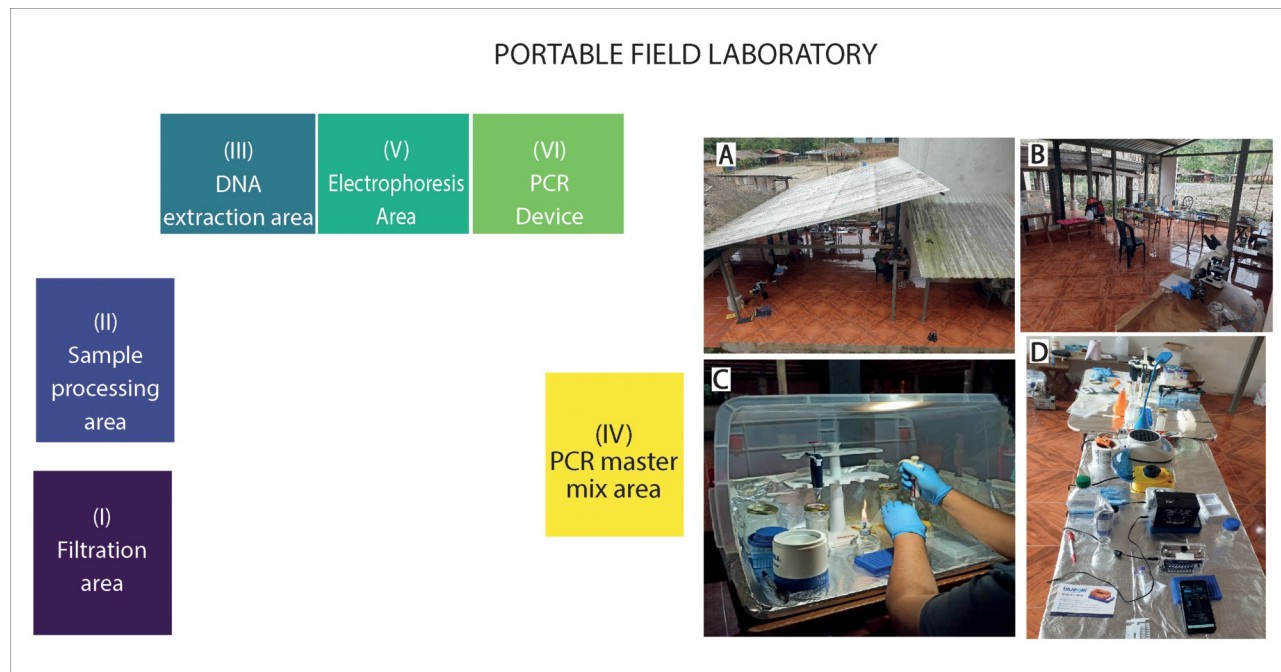

**Fig 1. A detailed view of the field laboratory is presented.** Numbers I, II, III, IV, V, and VI indicate the distribution of the various areas. Photo A shows the location in the field where the laboratory was set up. Photo B provides a general view of the laboratory, displaying all the areas. Photo C focuses on the PCR master mix area, while Photo D shows the table where the PCR mobile device, electrophoresis area, and DNA extraction areas were arranged.

discarded, and the remaining droplets were removed with a clean, sterile micropipette tip after spinning. The microtube was subsequently dried in a thermoblock at 60˚C until no liquid residue was observed. Finally, the DNA was resuspended in 25 μL of sterile MilliQ water. Due to logistical limitations, swab samples were only available for processing in the field laboratory; consequently, our results show the results obtained in the field laboratory.

For the extraction of DNA from the first-half nitrocellulose snippets in the field laboratory, we applied an adapted version of the Epicenter Masterpure DNA Purification Kit used by Riascos *et al*. [35]. The adaptations were the use of two sterile glass beads to the 1.5 ml tubes after homogenizing the samples, and changing the incubation time with proteinase K to thirty minutes and mixing by vortex every 10 minutes. The incubation time with RNAse at 37˚C was decreased to 10 minutes, and 175 μL of 5M ammonium acetate was added. The obtained DNA pellet was resuspended in 40 μL sterile demi-water instead of Epicentre's TE buffer.

DNA integrity was assessed in a blueGel electrophoresis system in a 0.8% agarose gel with 1X TBE and Gel Star™ Nucleic Acid Stain diluted 1:10,000 (Lonza, Lonza Rockland). For the identification of Bd in water and skin samples, we used the specific primers Chytr ITS1-3 (5′-CCTTGATATAATACAGTGTGC- CATATGTC-3′) and Chytr—5.8S (5′-AGCCAAGAGATCCGTTGTCAA-3′) [36]. Each PCR reaction contained 7.5μL of sterilized MilliQ $H_2O$, 1μL of each primer at a concentration of 10mM, 12.5μL of TaqMan Environmental Master Mix 2.0 (Applied Biosystems, Cat. # 4396838), and 3μL of DNA for a total volume of 25μL per reaction. Thermal cycling was done in a mini8 thermal cycler (miniPCR bio) programmed as follows: initial denaturation at 95˚C for 120s, 35 cycles of denaturation at 95˚C for 60s, annealing at 60˚C for 30s, extension at 72˚C for 30s and final extension at 72˚C for 5 minutes. Electrophoresis and visualization of PCR products were performed in a blueGel electrophoresis system (miniPCR Bio; https://www.minipcr.com/) using a 2% agarose gel in 1x

Tris/Borate/EDTA solution (TBE buffer) with GelStar™ Nucleic Acid Stain diluted 1:10,000 (10,000X in DMSO) (Lonza, Lonza Rockland).

DNA extraction from the second halves of the filters was performed at the university laboratory with the same DNA extraction and PCR protocols used in the field. PCR products were examined by electrophoresis in a 2% agarose gel with 1X TBE for 25 min at an electric potential difference of 135 V. After electrophoresis, the gel was stained with GelStar™ Nucleic Acid Stain diluted 1:10000 (Lonza, Lonza Rockland). If samples from skin swabs or water showed a 146 bp band, the amplification was considered successful for Bd DNA.

### Sequencing products

To evaluate the reliability and accuracy of Bd detection, we randomly selected samples with amplified PCR products for Sanger sequencing. Positive amplicons were sent for sequencing to Macrogen facilities (Seoul, Korea) to confirm the sequences of the Bd nuclear ribosomal region. Sequences were manually trimmed and cleaned in Geneious Prime® (ver. 2020.2.2). Sequencing and BLAST were used to confirm that the expected Bd sequence had been amplified (Update date: 2022/10/14, See S2 File). Accession numbers of sequences are detailed in S1 Table (*SUB12413894, provisional). The prevalence was calculated by dividing the number of positive Bd samples by the total number of samples at each site.

## Results

### Detection of Bd from swabs in *Atelopus* species

During the monitoring, twelve individuals of *Atelopus* were founded and swabbed. Of these, ten tested positive for Bd (see Table 1). At the Cerro Negro site, the two specimens of *A. bomolochos* detected were positive for Bd. In the Cajas National Park at the Angas site, two out of three *A. nanay* specimens were positive for Bd, including one deceased specimen (HMOA 2399). This specimen found under a rock in a stream, exhibited lesions consistent with chytridiomycosis. Finally, for the Cerro de Hayas and Estero Arenas sites, six of the seven *A. balios* specimens were confirmed positive for Bd.

### Presence of Bd in water samples per site

In the field laboratory, the first half of the nitrocellulose snippets from the twelve samples were filtered, extracted and amplified. From these, six samples showed a positive signal for Bd. This procedure was replicated for the second halves in the university laboratory, where the positive and negative results obtained in the field were confirmed (see electrophoresis results in Fig 2 and Table 2). The results from the field and university laboratories were further confirmed by Sanger sequencing, with the positive amplicons, showed 100% identity with Bd.

**Table 1. *Atelopus* species detection and Bd infection rates from amphibian swabs collected during the monitoring campaign in central Andes and coastal Ecuador.**

| Code | Field site | *Atelopus* species | # individuals found | # Testing positive for Bd |
|------|------------|--------------------|--------------------|--------------------------|
| A | Cajas National Park/Sig Sig-Angas | *A. nanay* | 3 | 2 |
| CN | Cerro Negro | *A. bomolochos* | 2 | 2 |
| CH | Cerro de Hayas | *A. balios* | 5 | 4 |
| EA | Estero Arenas | *A. balios* | 2 | 2 |

# = number

* Bd = *Batrachochytrium dendrobatidis.*

## (A) Results from the field laboratory

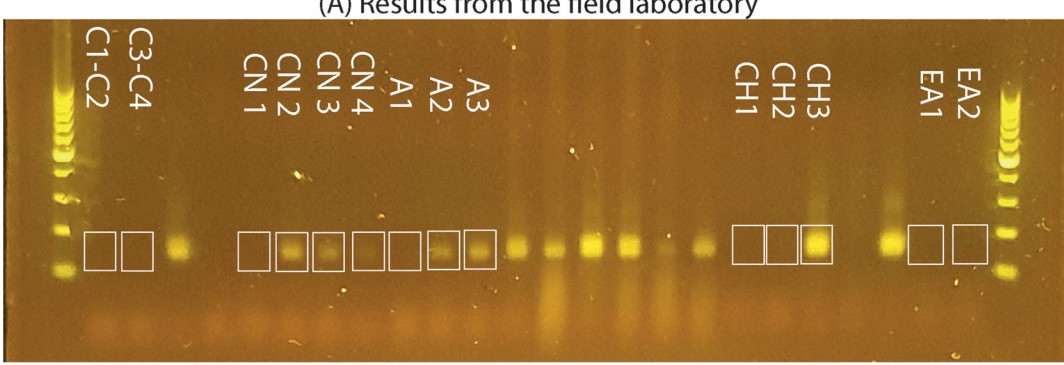

## (B) Results from the university laboratory

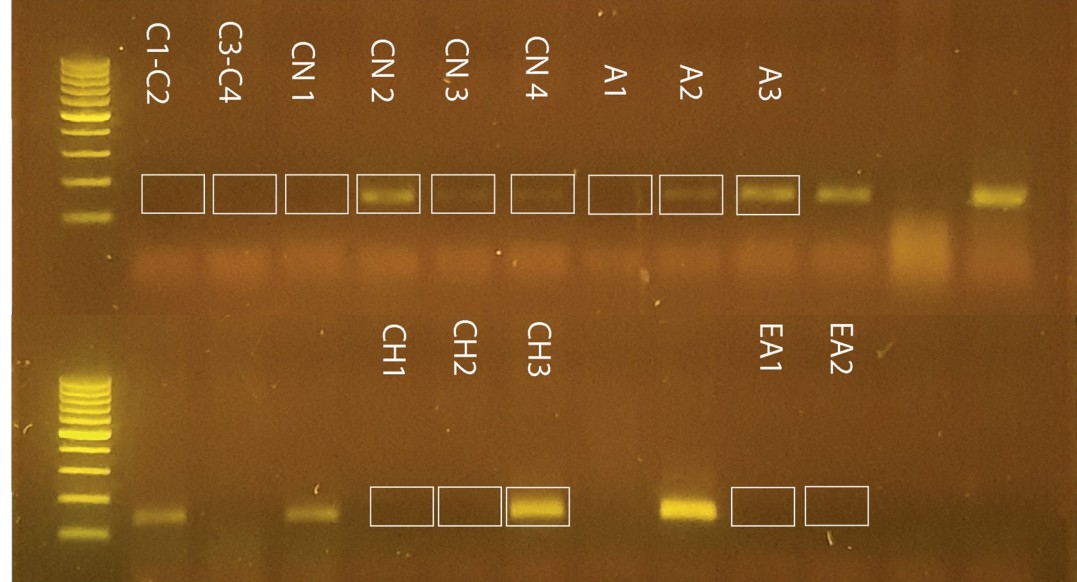

C : Negative control; CH: Cerro Haya location; CN Cerro Negro location; EA Estero Arena location; A Sig-Sig-Angas

**Fig 2.** Electrophoresis results of eDNA water samples processed (A) in the field and (B) in the laboratory.

The highest prevalence of Bd was found in the highlands around 3,000 m.a.s.l. where two out of the three samples collected in the Cajas National Park (location Sig-Sig-Angas) tested positive (See Table 1). Additionally, Bd was present in three out of the four sites in Cerro Negro. In both sites, not human disturbances were observed. In contrast, locations at lower elevation (300 m.a.s.l) had lower Bd detection rates than the highlands. For the location, Cerro

**Table 2.** Field results of water samples and amphibian swabs from four locations in Central Andes of Ecuador.

| Code | Location Name | eDNA sites collected | Field lab Bd positives | University lab Bd positives | Concordance rate (%) |
|---|---|---|---|---|---|
| A | Cajas National Park/Sig Sig-Angas | 3 | 2 | 2 | 100% |
| CN | Cerro Negro | 4 | 3 | 3 | 100% |
| CH | Cerro de Hayas | 3 | 1 | 1 | 100% |
| EA | Estero Arenas | 2 | 0 | 0 | 100% |

\* Bd = *Batrachochytrium dendrobatidis.*

eDNA = environmental DNA

de Hayas Arena, one out of the three samples showed a positive amplification for Bd. While in the location Estero Arena, Bd was absent from the water two sample collected. At both coastal sites, human disturbance was observed, including the presence of crops and livestock near the monitoring sites.

## Discussion

### The endangered toads *Atelopus* and the field laboratory as a conservation research instrument

Here we report the results of using a field laboratory for the detection of Bd at four sites in the Ecuadorian Andes and Coastal lowlands. Our results demonstrate that a basic laboratory (i.e. PCR instead of qPCR) combined with our sample processing methods is accurate and rapid (6–8 hours including transportation of samples) for detecting Bd in three critically endangered Andean toads. This applies to remote area characterized by diverse environments and different levels of disturbance. Moreover these results indicate that a field laboratory is just as effective as university laboratories in detecting Bd from environmental samples using molecular tools.

The *Atelopus* frogs examined in this study are classified as critically endangered toads [9]. Before the recent rediscovery of *A. balios* and *A. bomolochos*, both species had last been recorded in 1995 and 1990, respectively [18, 32]. The locations sampled for *A. balios* and *A. nanay* represent the last known habitats where these species have been observed. In addition to Bd presence, other threats to these species were identified. For example, in Cerro de Hayas and Estero Arenas, deforestation, cattle grazing and agro-chemicals were noted during our monitoring.

The efficiency of Bd detection in the field and the consistency of results in a university laboratory, without compromising accuracy, suggest that field-based devices are effective tools for rapid field pathogen detection in threatened amphibians. This capability is crucial for the implementation of disease surveillance and control strategies. Consequently, future monitoring efforts could expand to include the study of the 44 species across 5 families that have tested positive for Bd in 10 out of the 24 provinces of Ecuador (S2 Table). This expansion could be complemented by ecosystem monitoring through water physical chemical sampling.

### Influence of environmental and methodological factors in PCR amplification for water samples

Although eDNA results were consistent between the field and university laboratories, there were instances where samples from certain locations tested negative for Bd. These results could be explained by various factors such as the absence of the target organism (Bd) at the site, low DNA concentration, PCR inhibition or the influence of abiotic and biotic factors [37–39]. For instance, the coastal areas sampled for this study, where a higher frequency of negative results in eDNA water samples was recorded, were characterized by the presence of traditional crops like cocoa and banana. These crops often involve significant use of pesticides and fertilizers [40, 41]. In contrast, the Andean sites, which exhibited fewer negative results, are protected areas with low human impact.

Given this observations, future studies should explore the influence of external factors, such as pesticides (i.e. dithiocarbamates [42]) and fertilizers, on the stability and degradation of eDNA in aquatic ecosystems for tropical countries, especially in relation to their potential role causing zoospore mortality, increasing host susceptibility, and damaging eDNA [43, 44].

## Needs, challenges, and advantages of field molecular laboratories in remote areas

Early detection of invasive or pathogenic species in the field with molecular accuracy is key for monitoring and surveillance for biodiversity conservation [45]. However, molecular techniques commonly used for pathogen detection, such as DNA extraction and PCR assays, often require laboratories with sophisticated and costly equipment. In this context, field laboratories and portable devices have emerged as promising tools for the rapid and accurate detection of emerging pathogens [46]. The advantages of these portable devices include their ease of transportation to sampling site, convenient handling and immediate, results [29, 47].

Filtration of water samples and eDNA extraction close to amphibian collection sites significantly reduce the time between sampling and filtration. Thereby increasing the potential to maximize eDNA recovery from fresh samples [48]. Furthermore, the use of portable equipment mitigates the influence of certain detrimental factors on the viability of eDNA samples. In particular, maintaining the cold chain (~ -4°C) during the transport of eDNA samples from remote localities to the laboratory is considered a challenge because high temperatures negatively affect DNA concentration [49]. For example, for this specific study in Ecuador, storing and transporting water samples for environmental DNA analysis to a research facility or university laboratory took two to three days by road. Other advantages include the use of basic standardized molecular techniques in low-cost equipment for DNA extraction and conventional PCR, which can overcome many of the problems associated with the scarcity of high-cost laboratory equipment in emerging economies, thus helping to increase awareness and access to modern, portable technology, further building and strengthening research capacity in developing countries [25].

In Ecuador, as in other tropical countries with developing economies, equipment and supplies from local suppliers can be up to two to five times higher than in the US or Europe [50]. This, couple with limited access to grants, inadequate budgets, and lack of laboratory infrastructure and equipment, hinders the monitoring of emerging pathogens' impact. For example, a typical national grant provides approximately US$ 5,000 to US$ 40,000 for total project costs (i.e. CEDIA https://cedia.edu.ec/beneficio/fondo-idi-universidades/financiamiento/), while the local cost for setting up a basic molecular laboratory ranges from US$ 10,000 to US$ 30,000. Consequently, about 80–90% of the project budget would need to be allocated to equipment purchase, limiting the budget allocation for sampling, reagents, and external services (e.g., Sanger sequencing), often rendering projects unfeasible.

The portable equipment used in our study, the miniPCR DNA Discovery System, which costs approximately US$ 950, includes a miniPCR mini16 thermal cycler, a blueGel electrophoresis system and a 2–20 μl micropipette [51]. This represents a lower purchase cost compared to standard university laboratory equipment. In addition, access to the product portfolio available abroad (reagents, equipment, laboratory supplies) is limited in Ecuador due to high import taxes, few distributors, and few available brands.

Globally, the impact of Bd on frog communities remains largely unstudied [9, 52, 53]. Although various strategies can be implemented to control the impact of Bd in the amphibians analysed, the first step requires a detailed identification of the areas and species affected. Portability, local availability and affordability were the main reasons for selecting the equipment used in this research. The implementation of low-cost technologies can facilitate sampling campaigns for pathogen or disease surveillance and become a fundamental tool for risk assessment and management [28], conservation of endangered species (e.g., treatment of infected individuals), and development of reintroduction or eradication plans (invasive species) [54, 55]. Therefore, it is crucial to apply methods to integrate the use of low-cost technologies in

the study of biological invasions, invasive species, emerging infectious diseases, and zoonoses [9] in the context of rapid and effective management of a lethal fungi that pose a global threat to diversity [56]. By leveraging the development of portable sequencers such as Minion ONT and improving protocols with cutting-edge biotechnology for real-time sequencing [28], we are helping to establish a field methods for monitoring of emerging diseases and threatened species as an integral part of future biodiversity genomics applied to conservation.

## Conclusions

This study underscores the critical importance of early and accurate detection of invasive or pathogenic species, such as Batrachochytrium dendrobatidis (Bd), in endangered amphibian populations. Our research demonstrates the viability of using portable, low-cost technologies for molecular detection in remote areas. This approach effectively addresses the logistical and resource challenges that are often encountered in developing countries, including Ecuador. Our finding indicate that field laboratories and portable devices are effective in detecting Bd in *Atelopus* toads and their aquatic ecosystems through eDNA analysis in a tropical context. The application of these tools enabled rapid pathogen detection, which is crucial for timely disease surveillance, risk assessment, and the formulation of conservation strategies.

## Supporting information

**S1 File. A standardized protocol to extract eDNA (water samples), DNA (SWABs), and amplifying PCR products to detect Bd with portable devices.**
(DOCX)

**S2 File. Sequences (.fasta) of positive Bd validated in BLASTn from SWABs (2395, 2397, 2420, 2421) and water (1a-16a) samples in the study area.** See S1 Table for specimens, access numbers in Genbank and water sample codes.
(DOCX)

**S1 Table. Locations for sampling swabs on *Atelopus* specimens and water from streams in the study area.** Detailed data on codes, access numbers to Genbank, GPS coordinates, altitude, temperature, and relative humidity (RH) are provided.
(XLSX)

**S2 Table. Literature review of amphibians with detected Bd in Ecuador.**
(DOCX)

## Acknowledgments

We thank Jackeline Arpi, Naty Aguilar, Evelyn Ocampos, Alex Arias, Walter Quilumbaquin and Noemí Torres for their support and assistance in field and laboratory work. We thank the reviewers whose comments, suggestions, and revisions improved the manuscript.

## Author Contributions

**Conceptualization:** Lenin R. Riascos-Flores, Leopoldo Naranjo-Briceño, Jomira K. Yánez-Galarza, Christine Van der Heyden, H. Mauricio Ortega-Andrade.

**Data curation:** Lenin R. Riascos-Flores, Julio Bonilla, Leopoldo Naranjo-Briceño, Katherine Apunte-Ramos, Grace C. Reyes-Ortega, Andrea Carrera-Gonzalez, Jomira K. Yánez-Galarza, Fausto Siavichay Pesántez, H. Mauricio Ortega-Andrade.

**Formal analysis:** Lenin R. Riascos-Flores, Julio Bonilla, Leopoldo Naranjo-Briceño, Katherine Apunte-Ramos, Grace C. Reyes-Ortega, Marcela Cabrera, Andrea Carrera-Gonzalez, Jomira K. Yánez-Galarza, H. Mauricio Ortega-Andrade.

**Funding acquisition:** Julio Bonilla, Fausto Siavichay Pesántez, Peter Goethals, Jorge Celi, Christine Van der Heyden, H. Mauricio Ortega-Andrade.

**Investigation:** Lenin R. Riascos-Flores, Leopoldo Naranjo-Briceño, Katherine Apunte-Ramos, Grace C. Reyes-Ortega, Marcela Cabrera, José F. Cáceres-Andrade, Andrea Carrera-Gonzalez, Jomira K. Yánez-Galarza, Fausto Siavichay Pesántez, Luis A. Oyagata-Cachimuel, Peter Goethals, Christine Van der Heyden, H. Mauricio Ortega-Andrade.

**Methodology:** Lenin R. Riascos-Flores, Leopoldo Naranjo-Briceño, Katherine Apunte-Ramos, Grace C. Reyes-Ortega, José F. Cáceres-Andrade, Andrea Carrera-Gonzalez, Jomira K. Yánez-Galarza, Peter Goethals, H. Mauricio Ortega-Andrade.

**Project administration:** Jorge Celi, Christine Van der Heyden, H. Mauricio Ortega-Andrade.

**Resources:** Julio Bonilla, Andrea Carrera-Gonzalez, Fausto Siavichay Pesántez, Christine Van der Heyden, H. Mauricio Ortega-Andrade.

**Software:** H. Mauricio Ortega-Andrade.

**Supervision:** Lenin R. Riascos-Flores, Leopoldo Naranjo-Briceño, Andrea Carrera-Gonzalez, Christine Van der Heyden, H. Mauricio Ortega-Andrade.

**Validation:** Lenin R. Riascos-Flores, Julio Bonilla, Leopoldo Naranjo-Briceño, Andrea Carrera-Gonzalez, Jomira K. Yánez-Galarza, Christine Van der Heyden, H. Mauricio Ortega-Andrade.

**Visualization:** Lenin R. Riascos-Flores, Leopoldo Naranjo-Briceño, Christine Van der Heyden, H. Mauricio Ortega-Andrade.

**Writing – original draft:** Lenin R. Riascos-Flores, Julio Bonilla, Leopoldo Naranjo-Briceño, Jomira K. Yánez-Galarza, Peter Goethals, Jorge Celi, Christine Van der Heyden, H. Mauricio Ortega-Andrade.

**Writing – review & editing:** Lenin R. Riascos-Flores, Julio Bonilla, Leopoldo Naranjo-Briceño, Katherine Apunte-Ramos, Grace C. Reyes-Ortega, Marcela Cabrera, José F. Cáceres-Andrade, Andrea Carrera-Gonzalez, Jomira K. Yánez-Galarza, Fausto Siavichay Pesántez, Luis A. Oyagata-Cachimuel, Peter Goethals, Jorge Celi, Christine Van der Heyden, H. Mauricio Ortega-Andrade.

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
