## [Decision Letter · Decision Letter 0]

11 May 2023

PONE-D-23-11480Portable field-laboratory for in-situ molecular detection of the lethal fungus Batrachochytrium dendrobatidis on three critically endangered Atelopus toad species in Ecuador.PLOS ONE

Dear Dr. Ortega-Andrade,

Thank you for submitting your manuscript to PLOS ONE. After careful consideration, we feel that it has merit but does not fully meet PLOS ONE’s publication criteria as it currently stands. Therefore, we invite you to submit a revised version of the manuscript that addresses the points raised during the review process.

We look forward to receiving your revised manuscript.

Kind regards,

José António Baptista Machado Soares, PhD

Academic Editor

PLOS ONE

Journal Requirements:

   "CVdH: “Networks 2019 phase 2 Ecuador biodiversity project”, the VLIR-UOS South Initiative ZEIN 2014Z15

CVdH, JC, JB, HMOA: “DNA-based monitoring for assessing the effect of invasive species on aquatic communities in the Amazon basin of Ecuador”, the VLIR-UOS South Initiative EC 2020 SIN 277B125 “Biomonitoring of aquatic environments in the Amazon using environmental DNA”

HMOA: “Conservation of Ecuadorian Amphibians and access to genetic resources-PARG” managed by the Ministry of Environment and Water of Ecuador;  Project “On the quest of the golden fleece in Amazonia: The first herpetological DNA - barcoding expedition to unexplored areas on the Napo watershed, Ecuador”, funded by the Secretaría Nacional de Ciencia y Tecnología del Ecuador (Senescyt-ENSAMBLE Grant #PIC-17-BENS-001), The World Academy of Sciences (TWAS Grant #16-095)"

3. We note that Figure 1 in your submission contain map/satellite images which may be copyrighted. All PLOS content is published under the Creative Commons Attribution License (CC BY 4.0), which means that the manuscript, images, and Supporting Information files will be freely available online, and any third party is permitted to access, download, copy, distribute, and use these materials in any way, even commercially, with proper attribution. For these reasons, we cannot publish previously copyrighted maps or satellite images created using proprietary data, such as Google software (Google Maps, Street View, and Earth). For more information, see our copyright guidelines: http://journals.plos.org/plosone/s/licenses-and-copyright.

Additional Editor Comments:

Dear authors,

I am pleased to say that most of the reviewers enjoyed the manuscript very much and we are excited about the possibility to publish your work. However, several different concerns were raised by them, and I agreed with them. For example, the original manuscript needs another level of refinement as there are many grammatical errors and incorrect statements throughout. It is also difficult to understand the figures as the quality is very poor and much of the text is unreadable. Finally, many of the citations are not quite matching the statements or are not the best choice. Another concern is how the data is shown and it would be recommended to compare to what can be obtained in the laboratory.

So, I kindly invite the authors to realize a thoughtful revision of the original version to achieve the improvement asked by the Reviewer and Plos ONE journal.

Thank you and best regards,

António Machado

Reviewers' comments:

Reviewer's Responses to Questions

**Comments to the Author**

1. Is the manuscript technically sound, and do the data support the conclusions?

Reviewer #1: Partly

Reviewer #2: No

Reviewer #3: Partly

Reviewer #4: Yes

Reviewer #5: Partly

2. Has the statistical analysis been performed appropriately and rigorously? 

Reviewer #1: I Don't Know

Reviewer #2: Yes

Reviewer #3: I Don't Know

Reviewer #4: N/A

Reviewer #5: Yes

3. Have the authors made all data underlying the findings in their manuscript fully available?

Reviewer #1: Yes

Reviewer #2: Yes

Reviewer #3: Yes

Reviewer #4: Yes

Reviewer #5: Yes

4. Is the manuscript presented in an intelligible fashion and written in standard English?

Reviewer #1: No

Reviewer #2: Yes

Reviewer #3: No

Reviewer #4: No

Reviewer #5: Yes

5. Review Comments to the Author

Reviewer #1: The work addresses an important issue and presents a methodology to detect this threat to amphibians in the field.

The title of the work is suggested to be adjusted to be in accordance with the objective and work carried out, since in reality the equipment is used in the field, but in reality I think the term laboratory is not appropriate

A title like this is proposed: "In-situ molecular detection of the lethal fungus Batrachochytrium dendrobatidis on three critically endangered Atelopus toad species in Ecuador"

On the other hand, another relevant point in the manuscript needs a review of the writing, since in some points there are repeated paragraphs. A language revision is suggested

Although their results show the efficiency of processing in the field, this is not compared to what can be obtained in the laboratory, it would be convenient to have these data.

In some parts of the manuscript there is more information than is necessary.

Review the comments in the manuscript, work needs minor revision.

Reviewer #2: The manuscript titled, “Portable field-laboratory for in-situ molecular detection of the lethal fungus Batrachochytrium dendrobatidis on three critically endangered Atelopus toad species in Ecuador” describes a field-deployable portable laboratory for detecting Batrachochytrium dendrobatidis from swab samples and eDNA samples. I was excited to see that researchers are continuing to evaluate field-based, molecular methods for the detection of organisms in real-time. However, the present manuscript appears only to be bringing molecular supplies from the laboratory to a field-site and not actually in the field where amphibians were captured. Further, although their set-up is technically portable, I couldn’t imagine a field crew bringing these supplies with them to the site where amphibians were captured and don’t believe that the authors adequately demonstrated the portability of this system. In short, although the research follows many of PLOS’s guidelines, I do not feel that the conclusions are supported by the studies experimental design or results. Therefore, the authors should consider lessening their claims of ‘portability’ and ‘field-deployable’ unless they show that this set-up could reasonably be brought directly to the environments where these animals are collected.

Minor Comments:

Line 42: Parasitizes instead of parasites.

Line 49: …A. bomolochos, as well as…

Line 124: Why are swabs (SWABS) capitalized throughout?

Line 132: Is this section repeated? If so, delete redundant section.

Line 298: I’m not sure I would consider 8 hours after collection ‘rapid’, and it is certainly not more rapid than the 8 hours it would take to do this in a laboratory setting.

Line 386: Where are these numbers coming from? I’d like a citation to support this claim.

Line 414: The authors did not present a direct price comparison between their proposed method and typical lab-based methods, so it is difficult to evaluate if their approach is truly “low-cost”. Further, as the authors only used less expensive versions of common lab machines (miniPCR), there is no reason to believe that these alternative machines are not already being used in lab-settings to return the same “low-cost” results.

Line 419: Are processing times in the field actually faster than if it would be performed in the field or have you just removed the time for transiting the samples by performing these assays in the field?

Line 425: Real-time sequencing was not performed in this study, so I am not sure how this statement fits into the present manuscript.

Figure 2: I disagree that this laboratory is particularly portable or field deployable. It appears that the authors only took laboratory supplies and moved them outdoors at a field research site.

Figure 2: DNA Extraction is mislabeled as VII when it should be III.

Figure 3: The image is blurry and difficult to assess. Further, I don’t see how a PC analysis is required for the demonstration of the portable kit.

Figure S1: The lane labels on the right of the bottom panel are incorrect as +1 and +2 do not appear on the bottom gel.

Figure S1: 12A did not amplify in the lab, but did in the field, though this discrepancy was not highlighted in the manuscript as an instance of false-positive in the field or false-negative in the lab. Similar for 9A, though one could argue there is a faint band in 9A top panel.

Reviewer #3: Bd is threatening amphibians worldwide. While active surveillance and long-term monitoring of infected amphibian populations and aquatic ecosystems is needed, researchers from developing countries are limited because of budget restraints, poor access to well-equipped lab facilities and lack of trained staff The authors focus their study on providing a reliable and accurate protocol for Bd molecular detection on genomic DNA from amphibian skin swabs and eDNA from water samples that allows rapid Bd detection at remote areas, at an affordable cost. For instance, they tested a protocol for in-situ detection of Bd on skin swabs from three critically endangered amphibian species from Ecuador, and eDNA from water samples. They collected both sample types at 4 different locations, and analyzed them in the field using a portable thermocycler, and at a standard laboratory. Their results show the presented protocol may be a valuable tool for in-situ Bd detection from skin swabs, however results from eDNA are not accurate.

The protocol tested in this study may be a valuable option for Bd surveillance and monitoring in endangered amphibian populations from Ecuador; unfortunately, there are several issues in the methods, results and discussion that make the manuscript difficult to understand, and need to be addressed with adequate explanations to confirm the validity of findings.

I would be happy to read a revised version of this manuscript and to see this information published.

MAJOR ISSUES:

I strongly recommend copy editing the manuscript so it would be presented in a more intelligible way.

The title should be rewritten as you are assessing the reliability of an in-situ Bd molecular detection protocol using a portable machine, and not the lab equipment per se. It could be: “In-situ molecular detection of the lethal fungus Bd on three critically endangered Atelopus toad species in Ecuador using a portable thermocycler.”

ABSTRACT:

Please change as needed after revising the manuscript.

Lines 54-55: is this true? I couldn’t find the results obtained in the standard (on-campus) lab. You need to include results from swabs.

Lines 59 -60: you need to mention this is only true for skin swab samples. According to your results you failed detecting Bd in eDNA from positive sites. You also need to address this in the discussion.

Keywords should be related to your study. For example: chytridiomycosis, molecular detection, disease surveillance, etc. See the annotated manuscript.

INTRODUCTION:

Included information concerning eDNA sampling for Bd detection and monitoring, and the advantages of in-situ Bd detection. You should mention previous studies that have used similar methodologies in other countries. Please check Kamoroff et al. 2022, it would help you organizing the whole manuscript.

METHODS:

Methods should be clear and better organized. Maybe you can describe all methods related to amphibian samples first, and then all concerning water samples. I don´t recall reading about the protocols used in the standard lab, so I don´t know if the field protocol was compared with the same protocol but performed in a controlled environment (standard lab) or if a completely different protocol (different extraction methods, qPCR, etc) was used there. This information should also be included in detail in the supplementary files (SM1) This is important for results interpretation and if someone wishes to reproduce your study.

You need to rewrite this section in a more detailed and clear way. Providing detailed protocols from both, field and campus labs, is essential.

- Why did you choose these three species? Please make clear that you knew Bd was previously detected at sampled sites and species. This is important because you are validating a method and you need positive sampled to do that.

- Why did you use an Environmental Risk Surface model and why it was important in this study? You need to say this.

-Delete lines 132 – 141 (you just copy pasted the prior lines).

- Lines 141-143: include this information within an ethical statement.

- Amphibian sampling: did you sample only adults? how did you collect amphibians (by hand, nets, traps, etc)? Did you change gloves between each individual? Were individuals held together Add references for how you sampled specimens. Were samples stored at room temperature, -4°C, -20°C? You can use Soto-Azat et al. 2013 as reference. Reorganize this section.

- Mention if equipment was disinfected within and between sites.

- There is no information concerning water samples collection in the water collection section, for example: sampled water bodies (in SM1 you mention rivers and lakes), distance between samples from the same water body, depth, etc. Also, you should include here the information provided in the first step from the protocol outlined in SM1.

- Line 175-182: I don´t recall reading about water physical chemical analysis in the introduction. Please include a brief explanation of why are you doing this, and why is it relevant for Bd detection. How and why did you select the 12 parameters? I see 13 parameters in SM3.

- Line 189: you are extracting fungal Genomic DNA. Mention if you followed the manufacturer protocol. If you made any modifications, they should be written in detail.

- Provide more details about the statistical analysis. I couldn’t find information on how Bd prevalence was calculated for amphibians and for streams. You also have to show how you determined the probability of detection, sensitivity and specificity of the tested protocol when compared to the standard lab protocol. Same for the PCA.

- You need to provide a brief description of the DNA extraction and PCR protocols used in the standard lab, and to include the whole protocols in the supplementary materials. This is crucial for understanding, interpreting and reproducing your work.

Line 217: It would be better to say “Bd molecular detection through conventional PCR”.

Line 226: you need to mention the manufacturer name. This is important for reproducibility.

Line 233 -235: This should be in the prior section. You mention you used a similar protocol in the standard lab. Please write the differences in detail. This is very important.

- How many replicates were analyzed per sample?

Line 242-244: This should be in the results section.

Line 248-255: did you include samples from skin swabs and eDNA from all sampled species and sites?

Line 254: Supplementary figure SF1 should be in the results.

- How did you selected the sampling sites and species? Where the studied populations known to be positive to Bd?

- You need to give a brief description of the lab (distribution, equipment, distance from sampling sites, etc).

RESULTS:

- This section needs to be better organized. You could use the same subheadings used in the methods section to make it easier to read and understand.

-You should provide information concerning the reliability of the tested protocol compared to the one used in the standard lab. Please include probability of detection, sensitivity and specificity of the field protocol compared to the gold standard or the ULAB assay.

- You need to report if results from field and standard labs were the same for skin swabs and water samples or if they failed to detect Bd in different technical samples from each site. I couldn´t find the results obtained in the standard lab. How many technical samples did you include per site? In Figures SF1 I see different replicate numbers per samples among sites. This should be standardized to allow proper interpretation of results.

- Line 262: this should be in the methods.

- Line 265: you should mention this in the methods. Specify if the specimen was necropsied and if it showed lesions consistent with chytridiomycosis.

- Line 270: I couldn´t find the information described here. Table S3 is about reported Bd positive amphibians from Ecuador.

- I did not find information about how long it took you to get the results in the field and on campus. The abstract said 8 hours, but it should be included in the results.

DISCUSSION:

- Line 295 – 300: to discuss the accuracy of detection you need to provide more information, starting from results obtained at the standard lab and statistical analysis (see above). We don´t know if your in-situ protocol is faster than the standard.

Line 314: change Bd amphibian pandemic for chytridiomycosis.

Line 315-318: this should be in the introduction.

Line 341-433: PCR inhibition can be discarded including an IPC in the samples analyzed in the stanrdad lab.

Line 344-345: What does this mean? Please discuss this.

Line 375: what do you mean by “short-term molecular techniques”?

Line 385: change “invasive species” to “emerging pathogens”.

- You need to discuss your findings: is the field protocol reliable and accurate compared to standard protocols? Is it faster than standard protocols? You also need to discuss the limitations of the study. For example, according to your results, your protocol allowed you to detect Bd from skin swabs in positive individuals, apparently being highly sensitive (there is not enough information about this); however, results from eDNA samples were ambiguous and apparently not reliable. You may recommend assessing the limit of detection (number of zoospores/sample) of your protocol from both type of samples in future studies.

CONCLUSIONS:

You don´t need to provide a summary in this section. Conclusions should say if your protocol is accurate and reliable or not. Did it allow you to detect Bd in positive samples from skin swabs and eDNA or not? Is it faster and cheaper than standard methods?

Line 415-417: according to your results you cannot rely on analysis from eDNA samples to assess Bd presence/absence at a site, apparently due to low sensitivity (further studies).

(i): You need to provide more information to make this conclusion (results from campus lab, time spent using both protocols). May be say this is true for skin swab samples, or provide information to support this conclusion.

(ii)This is a result, not a conclusion. Rewrite and use results from probability of detection to make conclusions about accuracy.

(iii) This is a fact, not a conclusion. You may say: The protocol presented here for in-situ Bd molecular detection using a portable thermocycler

MINOR ISSUES

- Please include references were needed (across the whole manuscript) to support or discuss your results. See the annotated file.

- You use “DNA extraction” or “DNA isolation” randomly across the manuscript. Please choose one and be consistent. I would rather use “DNA extraction”. The same occurs with “blank samples” and “negative controls”.

- See minor comments in the annotated file.

TABLES AND FIGURES

Please check all legends.

Table 1: You need to show the results from field and campus labs to allow for comparison.

Table ST1 could be better organized to make it easier to read.

Fig. 2: (II) Sample preparation area. (IV) Reagent preparation area. (VI) Amplification and product detection area. Numbers from the diagram do not match those in the legend.

Include arrows o show workflow. It should be unidirectional to avoid contamination.

F3: what do you mean by environmental variation?

SUPPLEMENTARY MATERIAL:

Please check the order of appearance matches the order of the material provided.

SM1

It would be better if you write the whole protocol in Spanish first, and then the whole protocol in English. As it is written now I causes confusion. In addition, you should also provide the outlined on-campus lab protocols to allow for comparison. This is extremely important for understanding and reproducing your work.

Protocol for water sample filtration: Step 1 should appear in the methods from the main document. When did you considered a river to be big or small? How far were the 3 sampling points? Was this distance the same at all water bodies despite size? You mention filters were stored at -20°C, but in the methods you said samples analyzed in the field were analyzed immediately after collection. Did you use the same protocol in the field?

I assume the equipment was disinfected between sites. Please make that clear in the protocol.

-Change “metagenomic” DNA for genomic “DNA”.

Protocolo de extracción de ADN genómico: dice muestras de raspado de piel. Favor cambiar por hisopado de piel.

The PCR and electrophoresis protocols are described only for skin swabs. I understood you use the same protocol for swabs and eDNA samples. Please make that clear in the headings, or if you used a different protocol, please write it down.

SM3

According to the methods section, ammonium was not included in the PCA.

ST1

I would recommend the authors to include data concerning PCR results from skin swabs and water samples in this table.

Please restructure the table legend. See minor comments in the annotated file.

Mention what does the asterisk in “Access number” means.

ST3

This table provides a nice summary of reported Bd cases in Ecuador, but is beyond the scope of this study. Please provide information concerning Bd detection in water samples from your study.

SF1

I see 4 samples from Cerro negro, 3 from Angas, 3 from Cerro Hayas and 2 from Estero Arenas. In the methods you said samples were divided in 3 parts and analyzed as 3 samples per site.

Where do samples 8a – 13a come from

Reviewer #4: This paper marks an important step forward in bringing molecular detection into the field. They do a great job of explaining the importance and significance of this study and report interesting results for Bd infection patterns of Atelopus in Ecuador. However, this manuscript needs another level of refinement as there are many grammatical errors and incorrect statements throughout. It is also difficult to understand the figures as the quality is very poor and much of the text is unreadable. Finally, many of the citations are not quite matching the statements or are not the best choice. Therefore, there are substantial changes that need to be made, after which I expect this paper would be acceptable.

Also, I would like the authors to add a section to the paper describing the physical set-up of the lab as it is shown in the photos in Figure 2. What precautions were taken to set up the physical space to avoid contamination. Where is this lab? What kind of facility? Some of this is described in Lines 185-187 but it would be nice to have a separate section dedicated to this as it is a key part of the study.

While I do not identify all the grammar errors present in this manuscript, below I highlight some specific parts of the manuscript that should be corrected:

Line 68-71: This sentence has many grammatical errors and is incorrect in stating the Bd is a zoonotic disease. Maybe you mean zoosporic?

Line 73-74: See this response to the Scheele et al. 2019 paper that provides convincing evidence that these specific numbers are not reliable. https://www.science.org/doi/10.1126/science.aay1838

Line 79-80: The statement that Bd “is spread around ecosystems mainly by human activities and vector species (e.g. fish, arthropods and crayfish)” is not actually known and is not supported by the citations listed. You could just cut the first two sentences of this paragraph and start with “Bd is transmitted via …”

Line 88: The best citation for qPCR of Bd is Boyle et al. 2004 doi:10.3354/dao060141

Line 181: It is unclear why the PCA was conducted and what it tells us.

Figure 1: The labels of the points are unreadable. Also, the inclusion of the ERS map does not seem necessary.

Figure 2: The low quality of this figure makes it hard to understand. Also there appear to be errors (for example the PCR machine box in green is labeled V but in the legend V says “electrophoresis area”). This is a critical part of the study so it should be a high quality figure. Also – do you mention anywhere that PCR prep was done in a clean hood/container, as shown in the photo labeled with IV?

Figure 3: This PCA is uninterpretable because of the low quality. Also, I don’t think the PCA results add much to your paper. Are you able to do a statistical test to determine how much each of the water parameters are influencing your differential results? Otherwise consider cutting this analysis.

I think that the supplementary figure showing the gels (SF1) should be a figure in the main text as it is a key result and shows important visual evidence of your findings. That being said, it should be clearly labeled with the samples and appropriate controls because it is very hard to understand as it is currently displayed.

Reviewer #5: I think the issue of neglected sampling areas due to prohibitive laboratory costs highlighted in this manuscript is an extremely important factor to address! I appreciate the demonstration of Bd detection via PCR rather than qPCR. I would have liked to see how long samples were stored/left at ambient temperature until DNA extraction in the portable lab.

Is there a reason you are capitalizing “SWABS” instead of using “swabs”?

Is there a reason you are capitalizing “BD” instead of using the standard format, “Bd”?

Rephrasing and other specific comments:

Line 42: consider changing “parasites” to “parasitizes”

Lines 68-69: consider rephrasing for clarity

Line 131: consider adding “National Red List” for readers unfamiliar with Ecuador’s National Red List vs IUCN Red List

Line 132: consider rephrasing for clarity

Lines 134-141 are a repeat of lines 124-132.

Lines 141-143: Consider rephrasing for clarity, for instance to, “We thank the Ministry of…”

Line 161: consider changing “subsequent” to “subsequently”

Line 181: consider rephrasing for clarity

Line 250: I’m not sure if “cloned” is an accurate word to use here.

Line 297: Consider removing “or sequencing”, since you mention on line 321 the need for field-based sequencing methods

Line 301: I am unsure of what is being said here. Consider removing or rephrasing for clarity.

Line 324: consider changing “equal” to something more precise, such as prevalence or number of positives. As a reader, it’s unclear exactly what “equal” means when the following sentence states that the negative results were different.

Lines 330-333: I would include a statement earlier here stating that coastal sites had more disturbance compared to Andean sites. As it reads now, I have trouble following the argument being made for water sample differences.

Line 346: what does “DOC” stand for?

Line 357: consider changing “The capacity to early detect invasive or…” to “The capacity for early detection of invasive or…”

Line 370: What is the “cold chain”? Consider defining this term

Figure 2: The numbering for DNA extraction area in the schematic (VII) does not match with the photo and the area labels (III). I am also unsure of what the bottom right photo is showing – consider adding a description or label.

Figure S1: The legends (sample names) don’t match with the gel photo. Why are there wells in the photo that are not labelled in the legend? For example, 8a in panel S1A has a band, but no sample name in the legend. Consider reformatting so that the legend only includes samples run on the gel.

I am not seeing positive controls on the S1B panel. I would think that positive controls would need to successfully amplify in both treatments to consider both treatments successful.

SM1: For the Spanish and English translations, consider putting all of the protocols in Spanish first, and all of the protocols in English second. As a reader who speaks only 1 language, it would be easier for to see all of the protocols together rather than skipping through to find each step in my language.

6. PLOS authors have the option to publish the peer review history of their article (what does this mean?). If published, this will include your full peer review and any attached files.

Reviewer #1: No

Reviewer #2: No

Reviewer #3: No

Reviewer #4: No

Reviewer #5: **Yes: **Imani Russell

---

## [Author Response · Author response to Decision Letter 0]

23 Aug 2023

Dear reviewers, 

Many thanks for your deep and detailed revision of our manuscript. We addressed all your suggestions and comments, responding specifically as Authors responds (AR). 

AR: Agree 

 "CVdH: “Networks 2019 phase 2 Ecuador biodiversity project”, the VLIR-UOS South Initiative ZEIN 2014Z15 

CVdH, JC, JB, HMOA: “DNA-based monitoring for assessing the effect of invasive species on aquatic communities in the Amazon basin of Ecuador”, the VLIR-UOS South Initiative EC 2020 SIN 277B125 “Biomonitoring of aquatic environments in the Amazon using environmental DNA” 

HMOA: “Conservation of Ecuadorian Amphibians and access to genetic resources-PARG” managed by the Ministry of Environment and Water of Ecuador; Project “On the quest of the golden fleece in Amazonia: The first herpetological DNA - barcoding expedition to unexplored areas on the Napo watershed, Ecuador”, funded by the Secretaría Nacional de Ciencia y Tecnología del Ecuador (Senescyt-ENSAMBLE Grant #PIC-17-BENS-001), The World Academy of Sciences (TWAS Grant #16-095)" 

AR: Agree. We update the financial disclosure and acknowledgments, as follows:

Acknowledgments

We thank Jackeline Arpi, Naty Aguilar, Evelyn Ocampos, Alex Arias, Walter Quilumbaquin and Noemí Torres for their support and assistance in field and laboratory work. 

Financial disclosure

CVdH: “Networks 2019 phase 2 Ecuador biodiversity project”, the VLIR-UOS South Initiative ZEIN 2014Z15.

CVdH, JC, JB, HMOA: “DNA-based monitoring for assessing the effect of invasive species on aquatic communities in the Amazon basin of Ecuador”, the VLIR-UOS South Initiative EC 2020 SIN 277B125 “Biomonitoring of aquatic environments in the Amazon using environmental DNA”

HMOA: “Conservation of Ecuadorian Amphibians and access to genetic resources-PARG” managed by the Ministry of Environment and Water of Ecuador; Project “On the quest of the golden fleece in Amazonia: The first herpetological DNA - barcoding expedition to unexplored areas on the Napo watershed, Ecuador”, funded by the Secretaría Nacional de Ciencia y Tecnología del Ecuador (Senescyt-ENSAMBLE Grant #PIC-17-BENS-001), The World Academy of Sciences (TWAS Grant #16-095)", Erasmus+ CBHE consortium “Nature-based living-lab for interdisciplinary practical and research semester on sustainable development and environmental protection in the Amazona Rainforest [NB-LAB]” (Grant number: 619346-EPP-12020-1-DE-EPPKA2-CBHE-JP). The funders had no role in study design, data collection and analysis, decision to publish, or preparation of the manuscript. 

3. We note that Figure 1 in your submission contain map/satellite images which may be copyrighted. All PLOS content is published under the Creative Commons Attribution License (CC BY 4.0), which means that the manuscript, images, and Supporting Information files will be freely available online, and any third party is permitted to access, download, copy, distribute, and use these materials in any way, even commercially, with proper attribution. For these reasons, we cannot publish previously copyrighted maps or satellite images created using proprietary data, such as Google software (Google Maps, Street View, and Earth). For more information, see our copyright guidelines: http://journals.plos.org/plosone/s/licenses-and-copyright. 

AR: We decide to delete the Figure which include imagery from the Google Earth and explain the distribution of the sites into the Methods, Study Area and Supplementary Table ST1. 

Review Comments to the Author  

Reviewer #1: 

The work addresses an important issue and presents a methodology to detect this threat to amphibians in the field. The title of the work is suggested to be adjusted to be in accordance with the objective and work carried out, since in reality the equipment is used in the field, but in reality, I think the term laboratory is not appropriate A title like this is proposed: "In-situ molecular detection of the lethal fungus Batrachochytrium dendrobatidis on three critically endangered Atelopus toad species in Ecuador". 

  

 On the other hand, another relevant point in the manuscript needs a review of the writing, since in some points there are repeated paragraphs. A language revision is suggested Although their results show the efficiency of processing in the field, this is not compared to what can be obtained in the laboratory, it would be convenient to have these data. In some parts of the manuscript there is more information than is necessary.  

  

Review the comments in the manuscript, work needs minor revision.  

Author Response (AR): We accepted the suggestion from the reviewer. The title has been adjusted to “In-situ molecular detection of the lethal fungus Batrachochytrium dendrobatidis on three critically endangered Atelopus toad species in Ecuador” 

Reviewer #2: 

The manuscript titled, “Portable field-laboratory for in-situ molecular detection of the lethal fungus Batrachochytrium dendrobatidis on three critically endangered Atelopus toad species in Ecuador” describes a field-deployable portable laboratory for detecting Batrachochytrium dendrobatidis from swab samples and eDNA samples. I was excited to see that researchers are continuing to evaluate field-based, molecular methods for the detection of organisms in real-time. However, the present manuscript appears only to be bringing molecular supplies from the laboratory to a field-site and not actually in the field where amphibians were captured. Further, although their set-up is technically portable, I couldn’t imagine a field crew bringing these supplies with them to the site where amphibians were captured and don’t believe that the authors adequately demonstrated the portability of this system. In short, although the research follows many of PLOS’s guidelines, I do not feel that the conclusions are supported by the studies experimental design or results. Therefore, the authors should consider lessening their claims of ‘portability’ and ‘field-deployable’ unless they show that this set-up could reasonably be brought directly to the environments where these animals are collected.  

  

Minor Comments:  

1. Line 42: Parasitizes instead of parasites.  

AR: Line 46 has been revised to use 'parasitizes' instead of 'parasites'  

2. Line 49: …A. bomolochos, as well as…  

AR: We included A. bomolochos, as well as .. in line 54  

3. Line 124: Why are swabs (SWABS) capitalized throughout?  

AR: We have now corrected the text to use 'swabs' in lowercase instead of 'SWABS'.  

4. Line 132: Is this section repeated? If so, delete redundant section.  

AR: We removed the repeated section.  

5. Line 298: I’m not sure I would consider 8 hours after collection ‘rapid’, and it is certainly not more rapid than the 8 hours it would take to do this in a laboratory setting.  

AR: The mobilization time from sampling sites to the portable lab ranged between 2 to 4 hours (Section: Materials and Methods, Study area) depending on the sampling zone, whilst the time to reach the University laboratory was about two to three days. Adequately equipped laboratory necessary to do this work are not present in most of the Ecuadorian Universities mainly to the high cost. Therefore, we propose a cost-effective alternative which offer both portability and the capacity to obtain reliable results in regions close to the remote monitoring sites. The mobile laboratory was packed in compact containers and transported in a 4x4 truck. This is better described in materials and methods, and discussed in the main section. 

6. Line 386: Where are these numbers coming from? I’d like a citation to support this claim. 

AR: We have included a cite in the text, related to “The reality of scientific research in Latin America; an insider’s perspective” to support this claim. We contextualize the limitations to grants, inadequate budgets, and lack of laboratory infrastructure and equipment contribute to the dearth of monitoring to assess the impact of invasive species in Ecuador (lines 391-408) 

7. Line 414: The authors did not present a direct price comparison between their proposed method and typical lab-based methods, so it is difficult to evaluate if their approach is truly “low-cost”. Further, as the authors only used less expensive versions of common lab machines (miniPCR), there is no reason to believe that these alternative machines are not already being used in lab-settings to return the same “low-cost” results. 

REALIZAR ESTA COMPARACIÓN E INCLUIR EN ANEXOS 

AR: We contextualize the limitations to grants, inadequate budgets, and lack of laboratory infrastructure and equipment contribute to the dearth of monitoring to assess the impact of invasive species in Ecuador (lines 391-408) 

8. Line 419: Are processing times in the field actually faster than if it would be performed in the field or have you just removed the time for transiting the samples by performing these assays in the field?  

AR: It has been rephrased to explain processing time is faster because it is done in the field without transiting the samples (lines 375-390). 

9. Line 425: Real-time sequencing was not performed in this study, so I am not sure how this statement fits into the present manuscript.  

AR: This has been corrected to state the application of new sequencing technologies, like Minion, combined with field laboratories can be beneficious. 

10. Figure 2: I disagree that this laboratory is particularly portable or field deployable. It appears that the authors only took laboratory supplies and moved them outdoors at a field research site. 

AR: We agree with the comment of the reviewer, then we change the term “portable” to in situ field-laboratory along the text. 

  

11. Figure 2: DNA Extraction is mislabeled as VII when it should be III.  

AR: Corrected 

  

12. Figure 3: The image is blurry and difficult to assess. Further, I don’t see how a PC analysis is required for the demonstration of the portable kit.  

AR: We improved the Figure 2. We also consider important, to consider the contribution of physical-chemical parameters to the environmental variation and clustering of sampled localities, because they have different human disturbances. We discuss this pattern in lines 333-363. 

  

13. Figure S1: The lane labels on the right of the bottom panel are incorrect as +1 and +2 do not appear on the bottom gel.  

AR:Corrected 

  

14. Figure S1: 12A did not amplify in the lab, but did in the field, though this discrepancy was not highlighted in the manuscript as an instance of false-positive in the field or false-negative in the lab. Similar for 9A, though one could argue there is a faint band in 9A top panel.  

AR: Those samples are not from this study. Corrected in the Figure S1. 

Reviewer #3: 

Bd is threatening amphibians worldwide. While active surveillance and long-term monitoring of infected amphibian populations and aquatic ecosystems is needed, researchers from developing countries are limited because of budget restraints, poor access to well-equipped lab facilities and lack of trained staff The authors focus their study on providing a reliable and accurate protocol for Bd molecular detection on genomic DNA from amphibian skin swabs and eDNA from water samples that allows rapid Bd detection at remote areas, at an affordable cost. For instance, they tested a protocol for in-situ detection of Bd on skin swabs from three critically endangered amphibian species from Ecuador, and eDNA from water samples. They collected both sample types at 4 different locations and analyzed them in the field using a portable thermocycler, and at a standard laboratory. Their results show the presented protocol may be a valuable tool for in-situ Bd detection from skin swabs, however results from eDNA are not accurate.  

  

The protocol tested in this study may be a valuable option for Bd surveillance and monitoring in endangered amphibian populations from Ecuador; unfortunately, there are several issues in the methods, results and discussion that make the manuscript difficult to understand, and need to be addressed with adequate explanations to confirm the validity of findings.  

  

I would be happy to read a revised version of this manuscript and to see this information published.  

  

MAJOR ISSUES: 

  

15. I strongly recommend copy editing the manuscript so it would be presented in a more intelligible way.  

AR: Corrected in the writing and editing style. 

16. The title should be rewritten as you are assessing the reliability of an in-situ Bd molecular detection protocol using a portable machine, and not the lab equipment per se. It could be: “In-situ molecular detection of the lethal fungus Bd on three critically endangered Atelopus toad species in Ecuador using a portable thermocycler.” 

AR: This has been addressed in the title and along the text. 

  

ABSTRACT:  

17. Please change as needed after revising the manuscript.  

Lines 54-55: is this true? I couldn’t find the results obtained in the standard (on-campus) lab. You need to include results from swabs.  

AR: This has been explained in table 1 and results section. 

18. Lines 59 -60: you need to mention this is only true for skin swab samples. According to your results you failed detecting Bd in eDNA from positive sites. You also need to address this in the discussion. 

AR: This has been explained in table 1 and results section. 

19. Keywords should be related to your study. For example: chytridiomycosis, molecular detection, disease surveillance, etc. See the annotated manuscript.  

AR: Keywords were changed. Appropriate keywords were used. 

  

INTRODUCTION:  

20. Included information concerning eDNA sampling for Bd detection and monitoring, and the advantages of in-situ Bd detection. You should mention previous studies that have used similar methodologies in other countries. Please check Kamoroff et al. 2022, it would help you organizing the whole manuscript.  

AR: Agreed. Thanks for this suggestion. 

  

METHODS:  

21. Methods should be clear and better organized. Maybe you can describe all methods related to amphibian samples first, and then all concerning water samples. I don´t recall reading about the protocols used in the standard lab, so I don´t know if the field protocol was compared with the same protocol but performed in a controlled environment (standard lab) or if a completely different protocol (different extraction methods, qPCR, etc) was used there. This information should also be included in detail in the supplementary files (SM1) This is important for results interpretation and if someone wishes to reproduce your study.  

AR: Improved, rewording and edited along the methods, results and discussion. Furthermore, SM1 was reviewed and corrected. 

22. You need to rewrite this section in a more detailed and clear way. Providing detailed protocols from both, field and campus labs, is essential.  

AR: Improved, rewording and edited along the methods, results and discussion. Furthermore, SM1 was reviewed and corrected. 

23. Why did you choose these three species? Please make clear that you knew Bd was previously detected at sampled sites and species. This is important because you are validating a method and you need positive sampled to do that.  

AR: We contextualize the selection of those species because in Ecuador we (the authors) were previously involved in the PARG project, which has targeted threatened species to develop a national conservation and biomonitoring program. Among them, the three species of Atelopus were selected to test the in situ molecular lab (lines 95-112). 

24. Why did you use an Environmental Risk Surface model and why it was important in this study? You need to say this.  

AR: We decide to delete this section and results, because consider that does not provide significant information in the context of the study. 

25. Delete lines 132 – 141 (you just copy pasted the prior lines).  

AR: Solved 

26. Lines 141-143: include this information within an ethical statement.  

AR: We include a section for Ethical statement (lines 130-134) 

27. Amphibian sampling: did you sample only adults? how did you collect amphibians (by hand, nets, traps, etc)? Did you change gloves between each individual? Were individuals held together Add references for how you sampled specimens. Were samples stored at room temperature, -4°C, -20°C? You can use Soto-Azat et al. 2013 as reference. Reorganize this section.  

AR: We include all the detailed information about methods for amphibian sampling, water collection and physical chemical analysis, and molecular processes (lines 164-265). 

27. Mention if equipment was disinfected within and between sites.  

AR. All material used during sampling was disinfected in the lab before arrival to sampling site. 

28. There is no information concerning water samples collection in the water collection section, for example: sampled water bodies (in SM1 you mention rivers and lakes), distance between samples from the same water body, depth, etc. Also, you should include here the information provided in the first step from the protocol outlined in SM1.  

AR: Clarified along the methods. 

29. Line 175-182: I don´t recall reading about water physical chemical analysis in the introduction. Please include a brief explanation of why are you doing this, and why is it relevant for Bd detection. How and why did you select the 12 parameters? I see 13 parameters in SM3.  

AR: Improved in the context of the introduction (lines 113-116). 

30. Line 189: you are extracting fungal Genomic DNA. Mention if you followed the manufacturer protocol. If you made any modifications, they should be written in detail.  

AR: We detail the complete protocol, which include modifications, in the SM1. 

31. Provide more details about the statistical analysis. I couldn’t find information on how Bd prevalence was calculated for amphibians and for streams. You also have to show how you determined the probability of detection, sensitivity and specificity of the tested protocol when compared to the standard lab protocol. Same for the PCA.  

AR: We detail how calculate prevalence in this study (lines 267-268). Furthermore, we describe in a better way the methods, results and discussion of the PCA analysis. 

32. You need to provide a brief description of the DNA extraction and PCR protocols used in the standard lab, and to include the whole protocols in the supplementary materials. This is crucial for understanding, interpreting and reproducing your work.  

AR: Clarified along the methods. We detail the complete protocol, which include modifications, in the SM1. 

33. Line 217: It would be better to say “Bd molecular detection through conventional PCR”.  

AR: Agree 

34. Line 226: you need to mention the manufacturer name. This is important for reproducibility.  

AR: Agree 

35. Line 233 -235: This should be in the prior section. You mention you used a similar protocol in the standard lab. Please write the differences in detail. This is very important. 

 AR: Agree, corrected. 

36. How many replicates were analyzed per sample?  

AR: We have three replicates per sample. We detailed it in methods. 

37. Line 242-244: This should be in the results section.  

AR: Agreed. 

38. Line 248-255: did you include samples from skin swabs and eDNA from all sampled species and sites?  

AR: We detailed it in methods. 

39. Line 254: Supplementary figure SF1 should be in the results.  

AR: We consider maintaining this figure as supplementary, because the same results area described in the corresponding section. 

40. How did you selected the sampling sites and species? Where the studied populations known to be positive to Bd?  

AR: We include a context in the introduction (lines 97-112) and methods (138-143). 

41. You need to give a brief description of the lab (distribution, equipment, distance from sampling sites, etc).  

AR: We improve the Figure 1 and describe in detail those aspects in the Materials and Methods section. 

  

RESULTS:  

42. This section needs to be better organized. You could use the same subheadings used in the methods section to make it easier to read and understand.  

AR: Agreed. 

43. You should provide information concerning the reliability of the tested protocol compared to the one used in the standard lab. Please include probability of detection, sensitivity and specificity of the field protocol compared to the gold standard or the ULAB assay.  

AR: We explain in detail the procedure and results of sampling and DNA processing. Also, we include as supplementary material those protocols, and discuss the differences between field-University laboratories (lines 343-352). 

44. You need to report if results from field and standard labs were the same for skin swabs and water samples or if they failed to detect Bd in different technical samples from each site. 

AR: We detailed it in methods that samples were from the same skin swabs and water samples. 

45. I couldn´t find the results obtained in the standard lab. How many technical samples did you include per site? In Figures SF1 I see different replicate numbers per samples among sites. 

AR: We detailed it in methods, for amphibian sampling and water collection. 

46. This should be standardized to allow proper interpretation of results.  

AR: We improved the parragraph structure and clarified all the suggestions from the reviewer. 

47. Line 262: this should be in the methods.  

AR: We maintain this lines in the results because is the number of specimens found and sampled. 

48. Line 265: you should mention this in the methods. Specify if the specimen was necropsied and if it showed lesions consistent with chytridiomycosis.  

AR: Agreed. 

49. Line 270: I couldn´t find the information described here. Table S3 is about reported Bd positive amphibians from Ecuador.  

AR: Corrected to Supplementary Table 1, and moved to Materials and Methods 

50. I did not find information about how long it took you to get the results in the field and on campus. The abstract said 8 hours, but it should be included in the results.  

AR: Agreed. We include information in lines 279-281. 

  

DISCUSSION:  

51. Line 295 – 300: to discuss the accuracy of detection you need to provide more information, starting from results obtained at the standard lab and statistical analysis (see above). We don´t know if your in-situ protocol is faster than the standard. 

AR: We explained in Materials and Methods about differences in time and results in field and University campus. 

52. Line 314: change Bd amphibian pandemic for chytridiomycosis.  

AR: OK. 

53. Line 315-318: this should be in the introduction.  

AR: Rewording in the discussion (lines 332-334). 

54. Line 341-433: PCR inhibition can be discarded including an IPC in the samples analyzed in the stanrdad lab.  

AR: agreed. 

55. Line 344-345: What does this mean? Please discuss this. 

AR: Discussed in lines 367-377 

56. Line 375: what do you mean by “short-term molecular techniques”?  

AR: We erase this phrase. 

57. Line 385: change “invasive species” to “emerging pathogens”.  

Ok. corrected 

58. You need to discuss your findings: is the field protocol reliable and accurate compared to standard protocols? Is it faster than standard protocols? You also need to discuss the limitations of the study. For example, according to your results, your protocol allowed you to detect Bd from skin swabs in positive individuals, apparently being highly sensitive (there is not enough information about this); however, results from eDNA samples were ambiguous and apparently not reliable. You may recommend assessing the limit of detection (number of zoospores/sample) of your protocol from both type of samples in future studies.  

AR: In this new version of the manuscript, we present an improvement in the description of Methods, Results and discussion to be considered by the reviewers. 

  

CONCLUSIONS:  

59. You don´t need to provide a summary in this section. Conclusions should say if your protocol is accurate and reliable or not. Did it allow you to detect Bd in positive samples from skin swabs and eDNA or not? Is it faster and cheaper than standard methods?  

Line 415-417: according to your results you cannot rely on analysis from eDNA samples to assess Bd presence/absence at a site, apparently due to low sensitivity (further studies).  

(i): You need to provide more information to make this conclusion (results from campus lab, time spent using both protocols). May be say this is true for skin swab samples, or provide information to support this conclusion.  

(ii)This is a result, not a conclusion. Rewrite and use results from probability of detection to make conclusions about accuracy.  

(iii) This is a fact, not a conclusion. You may say: The protocol presented here for in-situ Bd molecular detection using a portable thermocycler  

AR: In this new version of the manuscript, we present an improvement in the description of Methods, Results and discussion to be considered by the reviewers. 

  

MINOR ISSUES  

60. - Please include references were needed (across the whole manuscript) to support or discuss your results. See the annotated file.  

AR: Thanks for this suggestion. We double check the information and references cited into the text 

61. You use “DNA extraction” or “DNA isolation” randomly across the manuscript. Please choose one and be consistent. I would rather use “DNA extraction”. The same occurs with “blank samples” and “negative controls”.  

AR: Corrected along the text. 

- See minor comments in the annotated file.  

AR: Included in the new version. 

  

TABLES AND FIGURES  

62. Please check all legends.  

AR: Ok. 

63. Table 1: You need to show the results from field and campus labs to allow for comparison.  

AR: Corrected 

64. Table ST1 could be better organized to make it easier to read.  

AR: Corrected 

65. Fig. 2: (II) Sample preparation area. (IV) Reagent preparation area. (VI) Amplification and product detection area. Numbers from the diagram do not match those in the legend.  

Include arrows o show workflow. It should be unidirectional to avoid contamination.  

AR: Corrected 

66. F3: what do you mean by environmental variation?  

AR: Phrase erased. 

  

SUPPLEMENTARY MATERIAL:  

67. Please check the order of appearance matches the order of the material provided.  

SM1  

AR: Corrected 

68. It would be better if you write the whole protocol in Spanish first, and then the whole protocol in English. As it is written now I causes confusion. In addition, you should also provide the outlined on-campus lab protocols to allow for comparison. This is extremely important for understanding and reproducing your work.  

AR: Corrected 

69. Protocol for water sample filtration: Step 1 should appear in the methods from the main document. When did you considered a river to be big or small? How far were the 3 sampling points? Was this distance the same at all water bodies despite size? You mention filters were stored at -20°C, but in the methods you said samples analyzed in the field were analyzed immediately after collection. Did you use the same protocol in the field?  

AR: Corrected. In this new version of the manuscript, we present an improvement in the description of Methods, Results and discussion to be considered by the reviewers. 

70. I assume the equipment was disinfected between sites. Please make that clear in the protocol.  

AR: In this new version of the manuscript, we present an improvement in the description of Methods, Results and discussion to be considered by the reviewers. 

71. Change “metagenomic” DNA for genomic “DNA”.  Protocolo de extracción de ADN genómico: dice muestras de raspado de piel. Favor cambiar por hisopado de piel.  

AR: Corrected 

72. The PCR and electrophoresis protocols are described only for skin swabs. I understood you use the same protocol for swabs and eDNA samples. Please make that clear in the headings, or if you used a different protocol, please write it down.  

SM3  

AR: Corrected 

73. According to the methods section, ammonium was not included in the PCA.  

ST1  

AR: Corrected 

74. I would recommend the authors to include data concerning PCR results from skin swabs and water samples in this table.  

AR. We present the results in Table 1. 

75. Please restructure the table legend. See minor comments in the annotated file.  

Mention what does the asterisk in “Access number” means.  

AR: corrected. 

76. ST3 This table provides a nice summary of reported Bd cases in Ecuador, but is beyond the scope of this study. Please provide information concerning Bd detection in water samples from your study.  

AR: We include a line of context in the discussion about this table. 

77. SF1 I see 4 samples from Cerro negro, 3 from Angas, 3 from Cerro Hayas and 2 from Estero Arenas. In the methods you said samples were divided in 3 parts and analyzed as 3 samples per site.  Where do samples 8a – 13a come from 

AR: Corrected in Figure 2. Those samples correspond to other study. 

Reviewer #4: 

This paper marks an important step forward in bringing molecular detection into the field. They do a great job of explaining the importance and significance of this study and report interesting results for Bd infection patterns of Atelopus in Ecuador. However, this manuscript needs another level of refinement as there are many grammatical errors and incorrect statements throughout. It is also difficult to understand the figures as the quality is very poor and much of the text is unreadable. Finally, many of the citations are not quite matching the statements or are not the best choice. Therefore, there are substantial changes that need to be made, after which I expect this paper would be acceptable.  

 Also, I would like the authors to add a section to the paper describing the physical set-up of the lab as it is shown in the photos in Figure 2. What precautions were taken to set up the physical space to avoid contamination. Where is this lab? What kind of facility? Some of this is described in Lines 185-187 but it would be nice to have a separate section dedicated to this as it is a key part of the study.  

 While I do not identify all the grammar errors present in this manuscript, below I highlight some specific parts of the manuscript that should be corrected:  

  

78. Line 68-71: This sentence has many grammatical errors and is incorrect in stating the Bd is a zoonotic disease. Maybe you mean zoosporic?  

AR: Corrected 

  

79. Line 73-74: See this response to the Scheele et al. 2019 paper that provides convincing evidence that these specific numbers are not reliable. https://www.science.org/doi/10.1126/science.aay1838 

  

AR: Included in the introduction- 

80. Line 79-80: The statement that Bd “is spread around ecosystems mainly by human activities and vector species (e.g. fish, arthropods and crayfish)” is not actually known and is not supported by the citations listed. You could just cut the first two sentences of this paragraph and start with “Bd is transmitted via …”  

AR: Agreed. 

  

81. Line 88: The best citation for qPCR of Bd is Boyle et al. 2004 doi:10.3354/dao060141  

AR: Agreed. 

  

82. Line 181: It is unclear why the PCA was conducted and what it tells us.  

AR: In this new version of the manuscript, we present an improvement in the description of Methods, Results and discussion to be considered by the reviewers. 

  

83. Figure 1: The labels of the points are unreadable. Also, the inclusion of the ERS map does not seem necessary. 

AR: We decided to delete this figure because does not support substantial information to the paper. 

  

84. Figure 2: The low quality of this figure makes it hard to understand. Also there appear to be errors (for example the PCR machine box in green is labeled V but in the legend V says “electrophoresis area”). This is a critical part of the study so it should be a high quality figure. Also – do you mention anywhere that PCR prep was done in a clean hood/container, as shown in the photo labeled with IV?  

AR: Corrected and improved 

  

85. Figure 3: This PCA is uninterpretable because of the low quality. Also, I don’t think the PCA results add much to your paper. Are you able to do a statistical test to determine how much each of the water parameters are influencing your differential results? Otherwise consider cutting this analysis.  

AR: In this new version of the manuscript, we present an improvement in the description of Methods, Results and discussion to be considered by the reviewers. 

  

86. I think that the supplementary figure showing the gels (SF1) should be a figure in the main text as it is a key result and shows important visual evidence of your findings. That being said, it should be clearly labeled with the samples and appropriate controls because it is very hard to understand as it is currently displayed.  

AR: Agreed. 

Reviewer #5: 

I think the issue of neglected sampling areas due to prohibitive laboratory costs highlighted in this manuscript is an extremely important factor to address! I appreciate the demonstration of Bd detection via PCR rather than qPCR. I would have liked to see how long samples were stored/left at ambient temperature until DNA extraction in the portable lab.  

AR: In this new version of the manuscript, we present an improvement in the description of Methods, Results and discussion to be considered by the reviewers. 

  

87. Is there a reason you are capitalizing “SWABS” instead of using “swabs”?  

Is there a reason you are capitalizing “BD” instead of using the standard format, “Bd”?  

AR: corrected  

Rephrasing and other specific comments:  

88. Line 42: consider changing “parasites” to “parasitizes”  

AR: corrected  

89. Lines 68-69: consider rephrasing for clarity  

AR: corrected  

90. Line 131: consider adding “National Red List” for readers unfamiliar with Ecuador’s National Red List vs IUCN Red List  

AR: corrected  

91. Line 132: consider rephrasing for clarity Lines 134-141 are a repeat of lines 124-132.  

92. Lines 141-143: Consider rephrasing for clarity, for instance to, “We thank the Ministry of…”  

AR: corrected  

93. Line 161: consider changing “subsequent” to “subsequently”  

AR: corrected  

94. Line 181: consider rephrasing for clarity  

AR: corrected  

95. Line 250: I’m not sure if “cloned” is an accurate word to use here.  

AR: corrected  

96. Line 297: Consider removing “or sequencing”, since you mention on line 321 the need for field-based sequencing methods  

97. Line 301: I am unsure of what is being said here. Consider removing or rephrasing for clarity.  

AR: corrected  

98. Line 324: consider changing “equal” to something more precise, such as prevalence or number of positives. As a reader, it’s unclear exactly what “equal” means when the following sentence states that the negative results were different.  

AR: corrected  

99. Lines 330-333: I would include a statement earlier here stating that coastal sites had more disturbance compared to Andean sites. As it reads now, I have trouble following the argument being made for water sample differences. 

AR: corrected  

100. Line 346: what does “DOC” stand for?  

AR: corrected  

Line 357: consider changing “The capacity to early detect invasive or…” to “The capacity for early detection of invasive or…”  

AR: corrected  

101. Line 370: What is the “cold chain”? Consider defining this term  

AR: corrected  

  

102. Figure 2: The numbering for DNA extraction area in the schematic (VII) does not match with the photo and the area labels (III). I am also unsure of what the bottom right photo is showing – consider adding a description or label. 

AR: corrected  

  

103. Figure S1: The legends (sample names) don’t match with the gel photo. Why are there wells in the photo that are not labelled in the legend? For example, 8a in panel S1A has a band, but no sample name in the legend. Consider reformatting so that the legend only includes samples run on the gel.  

AR: corrected in Figure 2  

  

104. I am not seeing positive controls on the S1B panel. I would think that positive controls would need to successfully amplify in both treatments to consider both treatments successful.  

AR: corrected  

  

105. SM1: For the Spanish and English translations, consider putting all of the protocols in Spanish first, and all of the protocols in English second. As a reader who speaks only 1 language, it would be easier for to see all of the protocols together rather than skipping through to find each step in my language.  

AR: corrected

---

## [Decision Letter · Decision Letter 1]

4 Oct 2023

PONE-D-23-11480R1In situ molecular detection of the lethal fungus Batrachochytrium dendrobatidis on three critically endangered Atelopus toad species in Ecuador.PLOS ONE

Dear Dr. Ortega-Andrade,

Thank you for submitting your manuscript to PLOS ONE. After careful consideration, we feel that it has merit but does not fully meet PLOS ONE’s publication criteria as it currently stands. Therefore, we invite you to submit a revised version of the manuscript that addresses the points raised during the review process.

I am please to inform you that all reviewers agree that the revised manuscript is much better and improved. However, all reviewers recommend a futher improvement of the narrative and English Editing of the revised manuscript, as well as improvements in Figures and Table 1. Please carefully read all comments and properly reply all comments.

We look forward to receiving your revised manuscript.

Kind regards,

José António Baptista Machado Soares

Academic Editor

PLOS ONE

Journal Requirements:

Reviewers' comments:

Reviewer's Responses to Questions

**Comments to the Author**

1. If the authors have adequately addressed your comments raised in a previous round of review and you feel that this manuscript is now acceptable for publication, you may indicate that here to bypass the “Comments to the Author” section, enter your conflict of interest statement in the “Confidential to Editor” section, and submit your "Accept" recommendation.

Reviewer #1: All comments have been addressed

Reviewer #3: (No Response)

Reviewer #4: All comments have been addressed

2. Is the manuscript technically sound, and do the data support the conclusions?

Reviewer #1: Partly

Reviewer #3: Yes

Reviewer #4: Yes

3. Has the statistical analysis been performed appropriately and rigorously? 

Reviewer #1: N/A

Reviewer #3: I Don't Know

Reviewer #4: Yes

4. Have the authors made all data underlying the findings in their manuscript fully available?

Reviewer #1: Yes

Reviewer #3: Yes

Reviewer #4: No

5. Is the manuscript presented in an intelligible fashion and written in standard English?

Reviewer #1: Yes

Reviewer #3: Yes

Reviewer #4: No

6. Review Comments to the Author

Reviewer #1: The new version of the document took into consideration the comments from the first revision, generally in a satisfactory manner. The main problem was the title that is not in accordance with the objectives and results presented. In the new version, although some aspects were covered, the conclusions remain unclear what the achievements were.

Reviewer #3: Dear authors,

The manuscript has been considerably improved and it is now written in a more intelligible way. There are still some minor issues I would recommend you addressing before publication.

I strongly believe you should remove all the information related to physical-chemical analysis of water samples (including figure 3 and supplementary material), as it is not related with the aim of this study, and there is not a clear explanation of the relevance of the findings. Maybe dig a bit more in this in another publication. Also, it would be great if you can discuss the sensitivity of your protocol for Bd detection in water samples.

Please be consistent within the manuscript. Choose one term (e.g Bd vs chytrid pathogens; chytridiomycosis vs chytridium, in situ lab vs field lab, etc), and stick with it through the text. Otherwise it causes confusion. The same occurs with the time of samples analysis in situ. In the results section you say it takes 6-8 hours, and in the discussion you say it takes 4-6 hours.

Some information in the introduction, methods and discussion still need to be rephrased (see below).

Figures 1 and 2 are still blurry. There are still some inconsistencies between the names within figure 1 and the legend. In figure 2, I see some discrepancies between the results obtained in situ and in campus (samples 3a and 4a). See related comments and suggestions below.

Line 51: change “Bd in native populations…” to “Bd in wild populations… endangered native species”.

Line 56: you did not detect Bd prevalence. Please change to: “Bd was detected in…”

Line 60: monitoring of what? You need to say Bd detection (in situ Bd detection).

Line 61: please change “monitoring of a tropical invasive and zoosporic fungus” to “Bd”. It is better to keep it simple.

Line 72: change “mortal registered disease” to “mortal disease reported to date”. Change “produced” for “caused”. Also, Bd IS NOT a zoonotic fungus. Please change “zoonotic” for “chytrid”.

Line 74: change “After the” for “After a”.

Lines 78-83: please rephrase, it is confusing.

Line 81: change “chytridium” for chytridiomycosis. Please be consistent through the manuscript. Use Bd when talking about the pathogen, and chytridiomycosis when talking about the disease.

Line 91: add “amphibian” after “threatened”.

Line 97: change “Ecuador” for “the country”.

Lines 97-108: please provide similar information about the three species (distribution, Bd status). You say A. nanay lives only in the Cajas National Park, but you don´t say where the other 2 species are found. There is no information concerning Bd in A. balios.

Line 112: the mechanism of what? Do you mean the mechanism of infection? Please add as necessary.

Line 117: change “biomonitoring” for “pathogen surveillance” or “Bd surveillance”.

Line 121: change “detect in situ” for “detect the presence of Bd in situ in native…”

Line 128: change “security” for “biosafety”. This info should be in the sampling section.

Line 158: change “were” to “was”.

Lines 158-159: rephrase the swabbing method and cite.

Line 163: rephrase the first sentence.

Line164: change “arrival” for “arriving”, and “start” for “starting”.

Lines 172-181: this information differs from the protocol detailed in SM1. Also, you need to say if the first half of the filter was stored at room temperature, 4°C or other.

Lines 197-180: change “At the in situ…” to “To prevent contamination at the in situ laboratory, …”.

Line 200: change “Additional” to “ In addition,”.

Lines 262-264: the information concerning prevalence should be at the end of the previous section.

Line 263: change “is the number” for “ was calculated considering the number…”.

Lines 275-278: please rephrase, it is confusing.

Line 282: change “this” to “these”.

Line 286: change "show” to “showed”.

Line 293: change "ratesthan” to “rates than”

Line 296: you say you collected one water sample, but Table 1 says you collected wo samples and one was positive.

Line 329: change “with detected BD” to “ positive to Bd”.

Lines 331-332: The first sentence needs to be rephrased. It is confusing.

Line 334: change “pest management” to “disease surveillance”.

Line 335: delete “the amphibian”. Delete “epidemic for chytridiomycosis”. Add “in Ecuador” after “in situ”.

Line 338: delete “portable”.

Line 339: add “portable” after “and”. It should be “laboratories and portable molecular tools…”.

Line 343: were water samples results the same or similar? I thought they were identical, but after seeing figure 2 and the information presented in line 343, it appears not to be like that.

Line 347: change “PCR reaction” to “PCR inhibition”.

Line 348: add “sampled for this study” after “Coastal sites”.

Line: 349: change “pressures” to “activities”.

*I would delete the information presented in lines 348-356. You don´t have enough data to make this conclusions and it is not essential for this particular study.

Line 357: change “chytrid pathogens” for “Bd”. You are not studying other chytrid fungi.

Lines 360-370:delete this information. It is all true and interesting, but it is beyond the scope of this study.

Lines 376-378: Rephrase, it is confusing.

Lines 419-420: cite.

Lines 420-422: are you sure you want to say this? Maybe you can change “control the prevalence of Bd” to “control the impact of Bd”.

Line 424: change “sampling campaigns” to “ pathogen or disease surveillance”.

Line 430: change “of lethal” to “of a lethal”. Delete “and biological invasions”.

Line 438: change “Our main conclusions are:” to “Our study shows”.

* I would change conclusion (ii). The most important thing here is not the percentage of water samples that tested positive, but to mention and discuss (within the discussion section) the sensitivity of your protocol to detect Bd DNA in water samples.

SM1: revisar que los protocolos coincidan con la información descrita en métodos. Ej: en el Protocolo de filtración de muestras de agua dice: “Colectar 3 litros de agua”, mientras que en métodos dice: “One liter of water was taken for each site (about 350ml per point)”.

- 3.PROTOCOLO DE EXTRACCIÓN DE ADN METAGENÓMICO DE LOS FILTROS DE AGUA. - Cambiar ADN metagenómico por ADN ambiental.

- Dice: “cada filtro será cortado en 3 partes”, mientras que en los métodos dice: “filters were divided in two halves”.

The same occurs in the English version.

Reviewer #4: This manuscript is much improved, but still requires some editing for grammar and clarity. Additionally, I hope the quality of the figures can be improved because they are a bit hard to decipher, especially the text in the red box in Figure 1. I was not able to access the file listed as the supplement because of the unusual file type. Can this be provided as a .zip instead?

Here are a few line comments, but I believe the authors should do one more edit for grammar before resubmitting.

Line 71: “most infectious” and “most mortal” are not well-supported claims. I suggest tempering this language.

Line 81: “Chytridium” should be “Chytridiomycosis” and change “being Atelopus toads should be “and Atelopus are one of the most…”

Line 200 – Can you be more specific about which practices were carried out?

Line 343 – This sentence makes it sound like the eDNA results were different between the field and university labs, but the results said the results were identical. This is confusing.

Line 376 – Change this sentence to “Early detection of invasive or pathogenic species in the field with molecular accuracy is key for monitoring and surveillance for biodiversity conservation.”

Table 1 should also include the results from the on-campus lab for the eDNA samples. This can help clear up the confusion from Line 343.

7. PLOS authors have the option to publish the peer review history of their article (what does this mean?). If published, this will include your full peer review and any attached files.

Reviewer #1: No

Reviewer #3: No

Reviewer #4: No

---

## [Author Response · Author response to Decision Letter 1]

20 Dec 2023

Dear Editor,

We have an updated version of the reviewed manuscript “Field-Based molecular detection of Batrachochytrium dendrobatidis in critically endangered Atelopus toads and aquatic habitats in Ecuador -PONE-D-23-11480R1) submitted to PloSOne.

Our responses to reviewers are codified as AR.

Review Comments to the Author

Reviewer #1: The new version of the document took into consideration the comments from the first revision, generally in a satisfactory manner. The main problem was the title that is not in accordance with the objectives and results presented. In the new version, although some aspects were covered, the conclusions remain unclear what the achievements were.

AR: We appreciate your feedback. We have accordingly updated the text. The conclusions as well as other parts of the text have been completely modified.

Reviewer #3: Dear authors,

The manuscript has been considerably improved and it is now written in a more intelligible way. There are still some minor issues I would recommend you addressing before publication.

I strongly believe you should remove all the information related to physical-chemical analysis of water samples (including figure 3 and supplementary material), as it is not related with the aim of this study, and there is not a clear explanation of the relevance of the findings. Maybe dig a bit more in this in another publication. Also, it would be great if you can discuss the sensitivity of your protocol for Bd detection in water samples.

AR: Thank you for your input. Based on your suggestions, we decided to exclude the details concerning the physico-chemical parameters because consider this results are not directly related with the research question.

Please be consistent within the manuscript. Choose one term (e.g Bd vs chytrid pathogens; chytridiomycosis vs chytridium, in situ lab vs field lab, etc), and stick with it through the text. Otherwise it causes confusion. The same occurs with the time of samples analysis in situ. In the results section you say it takes 6-8 hours, and in the discussion you say it takes 4-6 hours.

AR: We have standardized the terminology used throughout our work. Additionally, we've provided more comprehensive explanations for specific terms and refined the description of the time intervals between sample transportation and extraction, clarifying the significance of these durations.

Some information in the introduction, methods and discussion still need to be rephrased (see below).

Figures 1 and 2 are still blurry. There are still some inconsistencies between the names within figure 1 and the legend. In figure 2, I see some discrepancies between the results obtained in situ and in campus (samples 3a and 4a). See related comments and suggestions below.

AC: The figures were changed; the inconsistencies were corrected.

Line 51: change “Bd in native populations…” to “Bd in wild populations… endangered native species”.

AR: The introduction has been revised, and the relevant section has been updated accordingly.

Line 56: you did not detect Bd prevalence. Please change to: “Bd was detected in…”

AR: The text has been corrected.

Line 60: monitoring of what? You need to say Bd detection (in situ Bd detection).

AR: This was corrected and re-worked. We substitute the in situ for field detection. 

Line 61: please change “monitoring of a tropical invasive and zoosporic fungus” to “Bd”. It is better to keep it simple.

AR: The text has been modified. 

Line 72: change “mortal registered disease” to “mortal disease reported to date”. Change “produced” for “caused”. Also, Bd IS NOT a zoonotic fungus. Please change “zoonotic” for “chytrid”.

AR: We redo these lines. 

Line 74: change “After the” for “After a”.

AR: The text has been corrected for “a” instead of “the”

Lines 78-83: please rephrase, it is confusing.

AR: The text has been rephrase.

Line 81: change “chytridium” for chytridiomycosis. Please be consistent through the manuscript. Use Bd when talking about the pathogen, and chytridiomycosis when talking about the disease.

AR: The text has been corrected and terminology has been checked

Line 91: add “amphibian” after “threatened”.

AR: The text has added

Line 97: change “Ecuador” for “the country”.

The text has been changed

Lines 97-108: please provide similar information about the three species (distribution, Bd status). You say A. nanay lives only in the Cajas National Park, but you don´t say where the other 2 species are found. There is no information concerning Bd in A. balios.

AR: A more detailed description was added in the Study area section

Line 112: the mechanism of what? Do you mean the mechanism of infection? Please add as necessary.

AR: The paragraph has been rewritten

Line 117: change “biomonitoring” for “pathogen surveillance” or “Bd surveillance”.

AR: The suggestion has been implemented

Line 121: change “detect in situ” for “detect the presence of Bd in situ in native…”

AR: The text has been modified. For instance in situ was modified by field laboratory

Line 128: change “security” for “biosafety”. This info should be in the sampling section.

AR: The text has transfer to the sampling section and the text has been rewritted 

Line 158: change “were” to “was”.

AR: The suggestion to replace "was" with "were" has been followed.

Lines 158-159: rephrase the swabbing method and cite.

AR: The text has been corrected and cited

Line 163: rephrase the first sentence.

AR: The first sentence was rephrase

Line164: change “arrival” for “arriving”, and “start” for “starting”.

AR: The text has been corrected

Lines 172-181: this information differs from the protocol detailed in SM1. Also, you need to say if the first half of the filter was stored at room temperature, 4°C or other.

AR: The information has been updated. The protocols provided in SM1 has been corrected.

Lines 197-180: change “At the in situ…” to “To prevent contamination at the in situ laboratory, …”.

AR: The changed has been applied “in situ” was substitute with “ in field” .

Line 200: change “Additional” to “ In addition,”.

AR: Agreed and corrected.

Lines 262-264: the information concerning prevalence should be at the end of the previous section.

AR: Corrected.

Line 263: change “is the number” for “ was calculated considering the number…”.

AR: Agreed and corrected.

Lines 275-278: please rephrase, it is confusing.

AR: The entire paragraph has been rephrased.

Line 282: change “this” to “these”.

AR: Agreed and corrected.

Line 286: change "show” to “showed”.

AR: Agreed and corrected.

Line 293: change "ratesthan” to “rates than”

AR: Agreed and corrected.

Line 296: you say you collected one water sample, but Table 1 says you collected wo samples and one was positive.

AR: Inconsistencies were corrected 

Line 329: change “with detected BD” to “ positive to Bd”.

AR: Agreed and corrected.

Lines 331-332: The first sentence needs to be rephrased. It is confusing.

AR: We redo part of the discussion section for instance this part was merged with other sections in discussion

Line 334: change “pest management” to “disease surveillance”.

AR: See previous comment.

Line 335: delete “the amphibian”. Delete “epidemic for chytridiomycosis”. Add “in Ecuador” after “in situ”.

AR: We redo part of the discussion section for instance this part was merged with other sections in discussion

Line 338: delete “portable”.

AR: See previous comment.

Line 339: add “portable” after “and”. It should be “laboratories and portable molecular tools…”.

AR: See previous comment.

Line 343: were water samples results the same or similar? I thought they were identical, but after seeing figure 2 and the information presented in line 343, it appears not to be like that.

AR: The water samples for field lab and university lab showed similar results, and these findings were corroborated by subsequent PCR reactions and sequencing.

AR: The discussion here implies that, at some sites, there was no amplification for Bd, even though some species collected nearby tested positive. 

Line 347: change “PCR reaction” to “PCR inhibition”.

AR: Agreed and corrected.

Line 348: add “sampled for this study” after “Coastal sites”.

AR: Agreed and corrected.

Line: 349: change “pressures” to “activities”.

AR: The information was rephrased

*I would delete the information presented in lines 348-356. You don´t have enough data to make thi*I would delete the information presented in lines 348-356. s conclusions and it is not essential for this particular study.

Line 357: change “chytrid pathogens” for “Bd”. You are not studying other chytrid fungi.

AR: Agreed and corrected.

Lines 360-370: delete this information. It is all true and interesting, but it is beyond the scope of this study.

AR: The information was deleted.

Lines 376-378: Rephrase, it is confusing.

AR: The lines have been rephrased.

Lines 419-420: cite.

AR: A citation was provided

Lines 420-422: are you sure you want to say this? Maybe you can change “control the prevalence of Bd” to “control the impact of Bd”.

AR: Agreed and corrected.

Line 424: change “sampling campaigns” to “ pathogen or disease surveillance”.

AR: Agreed and corrected.

Line 430: change “of lethal” to “of a lethal”. Delete “and biological invasions”.

AR: Agreed and corrected.

Line 438: change “Our main conclusions are:” to “Our study shows”.

* I would change conclusion (ii). The most important thing here is not the percentage of water samples that tested positive, but to mention and discuss (within the discussion section) the sensitivity of your protocol to detect Bd DNA in water samples.

AR: The conclusion was redo.

SM1: revisar que los protocolos coincidan con la información descrita en métodos. Ej: en el Protocolo de filtración de muestras de agua dice: “Colectar 3 litros de agua”, mientras que en métodos dice: “One liter of water was taken for each site (about 350ml per point)”.

AR:The method section was redo, mistake was due to cross between protocols in different MS 

- 3.PROTOCOLO DE EXTRACCIÓN DE ADN METAGENÓMICO DE LOS FILTROS DE AGUA. - Cambiar ADN metagenómico por ADN ambiental.

AR:The title was changed.

- Dice: “cada filtro será cortado en 3 partes”, mientras que en los métodos dice: “filters were divided in two halves”.

The same occurs in the English version.

AR: The error was corrected

Reviewer #4: This manuscript is much improved, but still requires some editing for grammar and clarity. Additionally, I hope the quality of the figures can be improved because they are a bit hard to decipher, especially the text in the red box in Figure 1. I was not able to access the file listed as the supplement because of the unusual file type. Can this be provided as a .zip instead?

Thanks for your comments we have adapted your suggestion to the MS. 

Here are a few line comments, but I believe the authors should do one more edit for grammar before resubmitting.

Line 71: “most infectious” and “most mortal” are not well-supported claims. I suggest tempering this language.

AR: The suggestion has been followed. 

Line 81: “Chytridium” should be “Chytridiomycosis” and change “being Atelopus toads should be “and Atelopus are one of the most…”

AR: The paragraph was edited

Line 200 – Can you be more specific about which practices were carried out?

AR: The paragraph was redo however this specific practices are now detailed in the MS and in the attachment section.

Line 343 – This sentence makes it sound like the eDNA results were different between the field and university labs, but the results said the results were identical. This is confusing.

AR: We redo the sentence. Here we referred to the no amplification of DNA from specific sites which can be due the physical chemical variables or influenced by human activities.

Line 376 – Change this sentence to “Early detection of invasive or pathogenic species in the field with molecular accuracy is key for monitoring and surveillance for biodiversity conservation.”

AR: The phrase was changed 

Table 1 should also include the results from the on-campus lab for the eDNA samples. This can help clear up the confusion from Line 343.

AR: Table 1 was changed

---

## [Decision Letter · Decision Letter 2]

8 Feb 2024

Field-Based molecular detection of Batrachochytrium dendrobatidis in critically endangered Atelopus toads and aquatic habitats in Ecuador

PONE-D-23-11480R2

Dear authors,

I am pleased to inform you that both reviewers enjoyed the manuscript very much and endorsed the revised manuscript for publication.

Thank you for choosing Plos ONE journal to publish your study.

Best regards,

António Machado

Reviewers' comments:

Reviewer's Responses to Questions

**Comments to the Author**

1. If the authors have adequately addressed your comments raised in a previous round of review and you feel that this manuscript is now acceptable for publication, you may indicate that here to bypass the “Comments to the Author” section, enter your conflict of interest statement in the “Confidential to Editor” section, and submit your "Accept" recommendation.

Reviewer #1: All comments have been addressed

Reviewer #6: All comments have been addressed

2. Is the manuscript technically sound, and do the data support the conclusions?

Reviewer #1: Yes

Reviewer #6: Yes

3. Has the statistical analysis been performed appropriately and rigorously? 

Reviewer #1: N/A

Reviewer #6: Yes

4. Have the authors made all data underlying the findings in their manuscript fully available?

Reviewer #1: Yes

Reviewer #6: Yes

5. Is the manuscript presented in an intelligible fashion and written in standard English?

Reviewer #1: No

Reviewer #6: Yes

6. Review Comments to the Author

Reviewer #1: The manuscript has been considerably improved and it is now written in a more intelligible way. There are still some minor issues I would recommend you addressing before publication.

Reviewer #6: It is a very interesting work because of the urgency of research on this pathogen, especially for herpetologists.

Good job!

7. PLOS authors have the option to publish the peer review history of their article (what does this mean?). If published, this will include your full peer review and any attached files.

Reviewer #1: No

Reviewer #6: **Yes: **Fausto Cabezas Mera

---

## [Editor Report · Acceptance letter]

6 Mar 2024

PONE-D-23-11480R2 

PLOS ONE

Dear Dr. Ortega-Andrade, 

I'm pleased to inform you that your manuscript has been deemed suitable for publication in PLOS ONE. Congratulations! Your manuscript is now being handed over to our production team.

Kind regards, 

on behalf of

Dr. António Machado 

Academic Editor

PLOS ONE